# Value Gradient weighted Model-Based Reinforcement Learning

**Claas A. Voelcker**[1,2], **Victor Liao**[1,3], **Animesh Garg**[1,2,4], **Amir-massoud Farahmand**[1,2]
[1] Vector Institute, [2] University of Toronto, [3] University of Waterloo, [4] Nvidia
Correspondence to `c.voelcker@cs.toronto.edu`

## Abstract

Model-based reinforcement learning (MBRL) is a sample efficient technique to obtain control policies, yet unavoidable modeling errors often lead to performance deterioration. The model in MBRL is often solely fitted to reconstruct dynamics, state observations in particular, while the impact of model error on the policy is not captured by the training objective. This leads to a mismatch between the intended goal of MBRL, enabling good policy and value learning, and the target of the loss function employed in practice, future state prediction. Naive intuition suggests that value-aware model learning would fix this problem and, indeed, several solutions to this objective mismatch problem have been proposed based on theoretical analysis. However, they tend to be inferior in practice to commonly used maximum likelihood (MLE) based approaches. In this paper we propose the Value-Gradient weighted Model loss (VaGraM), a novel method for value-aware model learning which improves the performance of MBRL in challenging settings, such as small model capacity and the presence of distracting state dimensions. We analyze both MLE and value-aware approaches and demonstrate how they fail to account for sample coverage and the behavior of function approximation when learning value-aware models. Fom this, we highlight the additional goals that must be met to stabilize optimization in the deep learning setting. To achieve this, we leverage the gradient of the empirical value function as a measure of the sensitivity of the RL algorithm to model errors. We verify our analysis by showing that our loss function is able to achieve high returns on the Mujoco benchmark suite while being more robust than maximum likelihood based approaches.

## 1 Introduction

Model-based Reinforcement Learning (MBRL) is a sample-efficient approach to obtain a policy for a given control problem. It solves the control optimization into two interleaved stages: model learning and planning. In the *model learning* stage, an approximate model of the environment is learned which is then utilized in the *planning* stage to generate new experience without having to query the original environment. Often this process is repeated by continuously updating the model with new experience and replanning based on the updated model. MBRL is an enticing paradigm for scenarios in which samples of the true environment are difficult or expensive to obtain, such as computationally intensive simulators, real-world robots, or environments involving humans, since the model can be used to generalize the policy to unseen regions of the state space. The approach has received a lot of attention and significant progress has been made in the field (Sutton, 1990; Deisenroth & Rasmussen, 2011; Levine & Koltun, 2013; Hafner et al., 2020; Moerland et al., 2020; Schrittwieser et al., 2020).

One of the core problems of model-based policy learning methods, however, is that the accuracy of the model directly influences the quality of the learned policy or plan (Schneider, 1997; Kearns & Singh, 2002; Ross & Bagnell, 2012; Talvitie, 2017; Luo et al., 2019; Janner et al., 2019). Model errors tend to accumulate over time, and therefore long-term planning under approximate models can lead to suboptimal policies compared to the performance achievable with model-free approaches. This is especially prevalent in settings with complex dynamics, such as robotic environments with discontinuities, which can be hard to model with common function approximation methods. These model approximation errors are nearly impossible to avoid with current methods due to limits of

function approximation in model and value learning algorithms, which cannot fully capture the full distribution over dynamics functions perfectly, and the use of finite datasets.

Hence, it is important that a model is accurate *where it counts* for the planning procedure, by modelling dimensions and data points that have a higher impact on the planning. But this objective is not captured in most current MBRL methods, which generally use maximum likelihood estimation (MLE) to learn a parametric model of the environment without involving information from the planning process. The misalignment between the *model learning* and *planning* stages of MBRL has recently received renewed interest and is now commonly termed the *objective mismatch* of reinforcement learning (Lambert et al., 2020), but the problems has been investigated in earlier works (Joseph et al., 2013). Several recent papers have investigated the objective mismatch (Abachi et al., 2020; Zhang et al., 2021; Ayoub et al., 2020; Grimm et al., 2020; 2021; Nikishin et al., 2022), but currently theoretical investigation and understanding of possible approaches do not perform well when applied to complex deep learning based approaches (Lovatto et al., 2020) or the proposed approaches rely on heuristics which might not be applicable generally (Nair et al., 2020).

**Summary of Contributions**. We present the *Value-Gradient weighted Model loss* (VaGraM) which rescales the mean squared error loss function with gradient information from the current value function estimate. We demonstrate the advantage of the VaGraM loss over previous approaches via the analysis of the optimization behavior of the Value-Aware Model Learning framework (Farahmand et al., 2017; Farahmand, 2018) and form two hypotheses for the lack of empirical performance gain despite theoretical intuition: (a) the theory does not account for the optimization trajectory induced by the loss function and (b) it also does not address how to counter problems that arise when the state-space is yet insufficiently explored in early stages of the model training. Our experiments show, qualitatively and quantitatively, that the VaGraM loss impacts the resulting state and value prediction accuracy, and that it solves the optimization problems of previously published approaches. Beyond pedagogical domains, we show that VaGraM performs on par with a current state-of-the art MBRL algorithms in more complex continuous control domains, while improving robustness to irrelevant dimensions in the state-space and smaller model sizes.

## 2 BACKGROUND

We consider the discounted MDP setting $(\mathcal{S}, \mathcal{A}, p, r, \gamma)$ (Puterman, 1994), where $\mathcal{S}$ denotes the state space, $\mathcal{A}$ the action space of an agent, $p$ is a transition probability kernel, $r : \mathcal{S} \times \mathcal{A} \to \mathbb{R}$ is a scalar reward function, and $\gamma$ denotes the reward discount factor. Following the standard setting of reinforcement learning, the goal is to obtain an agent which maximizes the reward function while interacting with the environment by taking actions after an optimal (potentially stochastic) policy $\pi^*$ without knowledge of the true transition kernel.

We will concentrate on value function-based methods to solve the reinforcement learning problem. With these, the aim is to learn a function $V_\pi : \mathcal{S} \to \mathbb{R}$ which represent the (discounted) reward obtained in state $s$ by following policy $\pi$ from there: $V_\pi(s) = \mathbb{E}_{(s_0, a_0, \dots)} \left[ \sum_{t=0}^{\infty} \gamma^t r(s_t, a_t) | s_0 = s \right]$. It is also helpful to define an action-value function $Q(s, a) = r(s, a) + \gamma \int p(s'|s, a)V(s')ds'$. Many approaches (Watkins & Dayan, 1992; Mnih et al., 2013; Wang et al., 2016; Haarnoja et al., 2018) try to learn this function by minimizing the deviations of the value function approximation to a bootstrap target: $\min_\phi \mathbb{E} \left[ (Q_\phi(s, a) - (r(s, a) + \gamma \int p(s'|s, a)V(s')ds'))^2 \right]$. This equation forms the core motivation for our investigation of MBRL.

### 2.1 MODEL-BASED REINFORCEMENT LEARNING

In the MBRL framework, an approximate model $\hat{p}$ is trained from data to represent the unknown transition function $p$. We will use the word 'model' to refer to the learned approximation and 'environment' to refer to the unknown MDP transition function.[1]

We concentrate on the Dyna algorithm (Sutton, 1990) and specifically investigate the impact of model errors on the planning procedure. Dyna uses a dataset $\mathcal{D}$ of past experiences from the envi-

---

[1]We limit the discussion in this paper to only the model while assuming that the reward function is either known or learned by mean squared error minimization. In all experiments, the reward function is learned using regression.

ronment $\mathcal{D} = (s_i, a_i, r_i, s_i')_{i=1}^N$. A parametric model $\hat{p}_\theta$ of the environment is learned by a maximum likelihood estimate using $\mathcal{D}$: $\theta^* = \arg\max_\theta \sum_{i=1}^N \log \hat{p}_\theta(s_i', r_i | s_i, a_i)$. This model $\hat{p}_\theta$ is then used to sample new next states $s_{\text{model}}' \sim \hat{p}_\theta(\cdot | s, a)$ to obtain better coverage of the state-action space. The samples are used to train the value function and policy as if they were samples from the environment. It is also possible to learn deterministic models, which we will denote as $f_\theta$ for clarity.

## 2.2 KEY INSIGHT: MODEL MISMATCH PROBLEM

One of the main drawbacks of model-based reinforcement learning is the fact that model errors propagate and compound when the model is used for planning (Schneider, 1997; Kearns & Singh, 2002; Talvitie, 2017). As a simple example, assume that a sample is collected from a deterministic model and has an error $\epsilon$. A value function based method will use the model sample to compute a biased bootstrap target $r(s, a) + \gamma V(s' + \epsilon)$.

The impact of the modelling error on the value function therefore depends on the size of the error and the local behavior of the value function. As an extreme example take a value function that only depends on a subset of all state observation dimensions. In this case, a large error in an irrelevant dimension has no consequence on the obtained policy, yet a maximum likelihood loss for the model cannot properly capture this behavior without prior handcrafted features.

We can motivate the use of MLE (such as the mean squared error for a Gaussian model with fixed variance) as a loss function by an upper bound: $\sup_{V \in \mathcal{F}} |\langle p - \hat{p}, V \rangle| \leq ||p - \hat{p}||_1 \sup_{V \in \mathcal{F}} ||V||_\infty \leq \sqrt{\text{KL}(p||\hat{p})} \sup_{V \in \mathcal{F}} ||V||_\infty$ (Farahmand et al., 2017), but this bound is loose and does not account for the geometry of the problem's value function. In our example above a mean squared error would penalize deviations equally by their $L_2$ norm without accounting for the relevance of the dimensions.

## 2.3 VALUE-AWARE MODEL LEARNING

To address the model mismatch, Farahmand et al. (2017) proposed *Value-aware Model Learning* (VAML), a loss function that captures the impact the model errors have on the one-step value estimation accuracy. The core idea behind VAML is to penalize a model prediction by the resulting difference in a value function. Given a distribution over the state-action space $\mu$ and a value function $V$, it is possible to define a value-aware loss function $\mathcal{L}_V(\hat{p}, p, \mu)$:

$$\mathcal{L}_V(\hat{p}, p, \mu) = \int \mu(s, a) \left| \overbrace{\int p(s'|s, a) V(s') \mathrm{d}s'}^{\text{environment value estimate}} - \overbrace{\int \hat{p}(s'|s, a) V(s') \mathrm{d}s'}^{\text{model value estimate}} \right|^2 \mathrm{d}(s, a) \tag{1}$$

and its empirical approximation $\hat{\mathcal{L}}_V$ based on a dataset $D = (s, a, s')_{i=1}^N$ of samples from $\mu$ and $p$:

$$\hat{\mathcal{L}}_V(\hat{p}, \mathcal{D}) = \sum_{(s_i, a_i, s_i') \in \mathcal{D}} \left| V(s_i') - \int (\hat{p}(s'|s_i, a_i)) V(s') ds' \right|^2. \tag{2}$$

It is worth noting that if the loss $\mathcal{L}_V$ is zero for a given model, environment and corresponding value function, then estimating the bootstrap target based on the model will result in the exact same update as if the environment were used. However, this is rarely the case in practice!

The main problem of this approach is that it relies on the value function, which is not known a priori while learning the model. In the original formulation by Farahmand et al. (2017), the value function is replaced with the supremum over a function space. While this works well in the case of linear value function spaces, finding a supremum for a function space parameterized by complex function approximators like neural networks is difficult. Furthermore, the supremum formulation is conservative and does not account for the fact that knowledge about the value function is gained over the course of exploration and optimization in a MBRL approach.

Instead of the supremum over a value function class, Farahmand (2018) introduced a modification of VAML called *Iterative Value-Aware Model Learning* (IterVAML), where the supremum is replaced with the current estimate of the value function, . In each iteration, the value function is updated based on the model, and the model is trained using the loss function based on the last iteration's value function. The author presents error bounds for both steps of the iteration, but did not test

the algorithm to ascertain whether the presented error bounds are sufficient to guarantee a strong algorithm in practice. Notably IterVAML provides an intuitive fix to the model-mismatch problem, yet overlooks two key optimization issues which lead to empirical ineffectiveness.

## 3 VALUE-GRADIENT WEIGHTED MODEL LOSS (VAGRAM)

We present Value-Gradient weighted Model loss (VaGraM), a loss which is value-aware and has stable optimization behavior even in challenging domains with function approximation. To motivate the loss function, we highlight two causes for the lack of empirical improvements of IterVAML over MLE based approaches. These phenomena are investigated and verified in detail in section 4.

**Value function evaluation outside of the empirical state-action distribution** IterVAML suffers when randomly initialized models predict next states that are far away from the current data distribution or if the optimization procedure leads the model's prediction outside of the covered state space. Since the value function has only been trained on the current data distribution, it will not have meaningful values at points outside of its training set. Nonetheless, these points can still achieve small value prediction errors if, due to the optimization process, the value function outside the training distribution happens to have the same value at the model prediction as at the environment sample. We therefore require that our value-aware loss function should not directly depend on the value function at the model prediction, since these might be potentially meaningless.

**Suboptimal local minima** Since the model can converge to a solution that is far away from the environment sample if the values are equal, we find that the model-based value prediction often performs poorly after updating the value function. We expect that the updated model loss forces the model prediction to a new solution, but due to the non-convex nature of the VAML loss, the model can get stuck or even diverge. This is especially prevalent when the previous minimum is situated outside of the empirically covered state space. A stable value-aware loss function should therefore have only one minimum in the state space[2] that lies within the empirical state distribution.

### 3.1 APPROXIMATING A VALUE-AWARE LOSS WITH THE VALUE FUNCTION GRADIENT

To derive a loss function that fulfils these requirements, we start from the assumption that the difference between the model prediction and the environment next states $s'$ are small. This is implicitly required by many MBRL approaches, since an MLE model cannot be used to estimate the next state's value otherwise. We also assume that the model has small transition noise, akin to the model assumptions underlying MSE regression, otherwise the difference between a model sample and the next state sample might be large. Under this assumption, the IterVAML loss can be approximated by a Taylor expansion of the value function, where we denote the expansion of $V$ around a reference point $s'$ as $\hat{V}_{s'}$ and obtain $\hat{V}_{s'}(s) \approx V(s') + (\nabla_s V(s)|_{s'})^\mathsf{T}(s - s')$. Using this expansion at the next state sample $s'_i \in \mathcal{D}$ collected from the environment for each tuple independently instead of the original value function, the VAML error can be stated as:

$$\hat{\mathcal{L}}_{\hat{V}} = \sum_{\{s_i, a_i, s'_i\} \in \mathcal{D}} \left( V(s'_i) - \int \hat{p}_\theta(s'|s_i, a_i)(V(s'_i) + (\nabla_s V(s)|_{s'_i})^\mathsf{T}(s' - s'_i)) \mathrm{d}s' \right)^2 \quad (3)$$

$$= \sum_{\{s_i, a_i, s'_i\} \in \mathcal{D}} \left( \int \hat{p}_\theta(s'|s_i, a_i) \left( (\nabla_s V(s)|_{s'_i})^\mathsf{T}(s' - s'_i) \right) \mathrm{d}s' \right)^2 \quad (4)$$

This objective function crucially does not depend on the value function at unknown state samples, all $s'_i$ are in the dataset the value function is trained on, which solves the first of our major problems with the VAML paradigm.

We can simplify the objective above even further if we restrict ourselves to deterministic models of the form $\hat{s}'_i = f_\theta(s, a)$. Since VAML requires the expectation of the value function under the model

---

[2]The full loss function will likely still admit additional local minima due to the non-linear nature of the model itself, but the global optimum should coincide with the true model and the loss function should be convex in the state space.

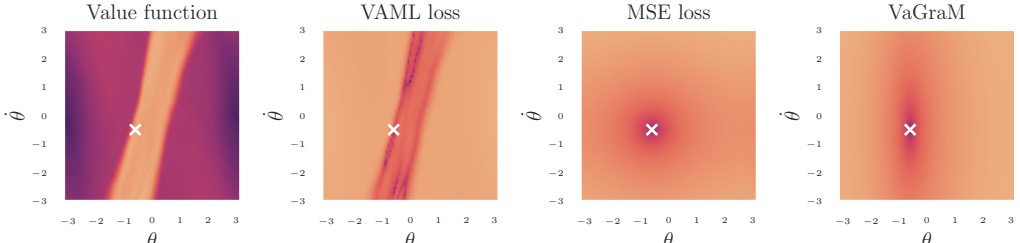

Figure 1: Visualization of discussed loss function with regards to a reference point marked with the white cross and the corresponding value function on the Pendulum environment. For the value function, darker color indicates a lower value. In the loss figures, darker color indicates how large the loss is if the model predicts $(\theta, \dot{\theta})$ instead of the reference sample marked in white. The VAML loss has a complex non-linear shape in the state space that follows isolines of the value function, while MSE and VaGraM are centered around the sample. For VaGraM, the rescaling of the MSE in the direction of high gradient along the $\theta$ axis is visible. Due to Equation 7, the scaling is aligned with the axis of the coordinate system and not rotated to fit the value function closer.

and the environment to be equal, we can exchange the probabilistic model with a deterministic one as long as we assume that the mean value function under the true environment is close to the empirical estimate of the value function from a single sample. We explore the prerequisites and consequences of this assumption further in Appendix F. The model loss can then be expressed as:

$$\sum_i \left( (\nabla_s V(s)|_{s_i'})^\intercal (f_\theta(s_i, a_i) - s_i') \right)^2 \tag{5}$$

We can see that the objective is similar to a mean squared error regression with a vector that defines the local geometry of the objective function. This vector can be interpreted as a measure of sensitivity of the value function at each data point and dimension. In regions where the value function changes significantly, the regression incentivizes the model to be very accurate.

### 3.2 Preventing spurious local minima

The formulation above retains one problem, Equation 5 does not constrain the solution for each $(s, a, s')$ tuple sufficiently. For each $(s, a, s')$ tuple, the loss function only requires that the difference between the model and environment sample be orthogonal to the gradient of the value function, which describes a hyperplane of solutions. These predictions can lie arbitrarily far away from the environment sample, which breaks the assumption underlying the Taylor approximation that the model prediction is within a small region of the expanded state point. For more details see Appendix A.

To prevent these suboptimal solutions and achieve our second design goal, we consider an upper bound on the value-gradient loss by applying the Cauchy Schwartz inequality $\left( \sum_{i=1}^n x_i \right)^2 \leq n \sum x_i^2$ to change the square of the sum with a sum of squares. We denote the diagonal matrix with vector $a$ on the diagonal as $\text{diag}(a)$ and refer to the dimensionality of the state space as $\dim(\mathcal{S})$ and rephrase the sum as a vector-matrix multiplication:

$$\sum_{\{s_i, a_i, s_i'\} \in \mathcal{D}} \left( (\nabla_s V(s)|_{s_i'})^\intercal (f_\theta(s_i, a_i) - s_i') \right)^2 \tag{6}$$

$$\leq \dim(\mathcal{S}) \sum_{\{s_i, a_i, s_i'\} \in \mathcal{D}} \left( (f_\theta(s_i, a_i) - s_i')^\intercal \text{diag}(\nabla_s V(s)|_{s_i'})^2 (f_\theta(s_i, a_i) - s_i') \right). \tag{7}$$

This reformulation is equivalent to a mean squared error loss function with a per-sample diagonal scaling matrix. Because the scaling matrix is positive semi-definite by design, each summand in the loss is a quadratic function with a single solution as long as the derivative of the value function does not become zero in any component. Therefore this upper bound assures our second requirement: the loss function does not admit spurious local minima.[3]

---

[3]We note that state dimensions are ignored for points in which components of the value function become zero, potentially leading to additional solutions, but in practice this rarely happens for more than a few points.

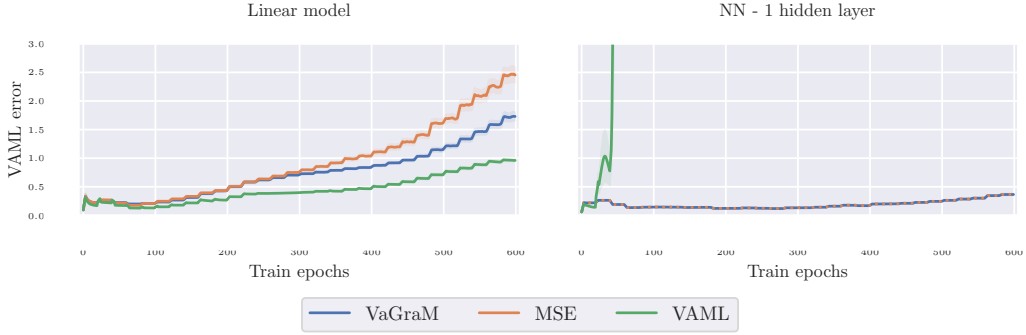

Figure 2: **Evolution of the VAML loss over changing value functions on the Pendulum domain**. Lines denote the mean and shaded areas show standard error over 8 model initialization and data set samples per model. In the linear setting, VAML achieves the lowest VAML error, while VaGraM is able to significantly outperform MSE. In the NN setting, VAML diverges rapidly, while VaGraM and MSE converge to approximately the same solution.

To give an intuitive insight into all the discussed loss functions, we visualized each one for a pedagogical environment, the Pendulum stabilization task. The resulting loss curves can be seen in Figure 1. The VAML loss has a complicated shape that depends on the exact values of the value function while both MSE and our proposal have a paraboloid shape. Compared to MSE, our proposed loss function is rescaled to account for the larger gradient of the value function in the $\theta$ axis.

## 4 EXPERIMENT: MODEL LEARNING IN LOW-DIMENSIONAL PROBLEM

We compare the performance of VaGraM, with both MSE and VAML on a pedagogical environment with a small state space and smooth dynamics to gain qualitative insight into the loss surfaces. We use the Pendulum environment, a canonical control problem in which an under-actuated pendulum must be swung and stabilized to an upright position. We use the implementation provided by Brockman et al. (2016). To learn the policy and its value function, we use the SAC algorithm (Haarnoja et al., 2018). The original IterVAML paper assumed that the value function was obtained using approximate value iteration (AVI) (Gordon, 1995; Ernst et al., 2005; Farahmand et al., 2010). We use SAC instead of a full AVI for stability in large scale experiments and discuss a proper extension of the VAML loss to SAC in Appendix C. We find that the difference in loss is negligible and therefore use SAC together with VAML throughout our experiments. More information on the implementation and hyperparameters of all of our experiments can be found in Appendix E.

To simplify the setup of the evaluation, we decided to investigate the model losses without model-based value function learning. This allows us to focus solely on the loss functions, without taking into account the inter-dependency between model and value function updates. Instead of the model-based loop, we used the SAC algorithm in a model-free setup to estimate the value function. We saved the intermediate value functions after each epoch of training, corresponding to 200 environment steps, and optimized the models using stochastic gradient descent on the respective loss function, updating the value function used for the loss every 1000 model training steps. As the MLE loss, we used the mean squared error which assumes a Gaussian model with fixed variance.

To compare the optimization, we used two architectures, a linear regression without feature transformations and a neural network with a single hidden layer and 16 neurons. We sampled a dataset uniformly over the whole state space and used the environment transition function to compute ground truth next state samples. Finally, we evaluated each models VAML error with regards to the current value function on a held out dataset and plotted the results in Figure 2.

**Dependency on untrained value function estimates.** The first cause for lacking empirical performance with VAML that we discussed in section 3 was that the algorithm can move outside of the covered state space region where the value function prediction is often meaningless.

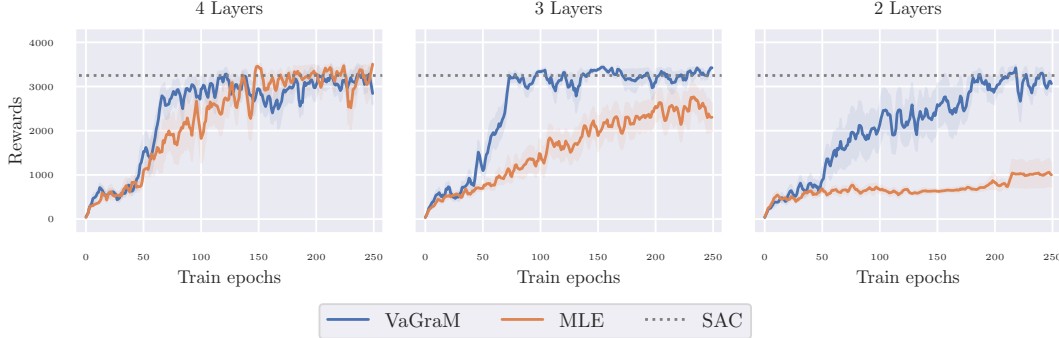

Figure 3: **Performance of VaGraM and MLE models with reduced model size**. The dotted lines correspond to the final performance reported for model-free SAC (grey, approx. 3200). Shaded area represents standard error over 16 repeated runs. VaGraM continues to solve the task almost unimpeded, while MLE is unable to even stabilize the Hopper when using a two layer neural network.

In our experiment we find that a linear regression model remains stable under all three loss functions, VaGraM, MSE and VAML. But when using a flexible function approximation, the VAML loss converges in the first iteration with the given value function, but then rapidly diverges once the value function is updated. When investigating the mean squared error of the VAML solution, we find that the model finds a stable minimum of the VAML loss outside of the reachable state space of the pendulum. This confirms our hypothesis that flexible VAML models can find solutions outside of the empirical state space distribution, which are unstable once we update the value function. VaGraM remains stable even with flexible function approximation and achieves a lower VAML error than the MSE baseline when using a model with insufficient capacity to represent the dynamics.

**Single solution convergence.** In the experiment, we see that the MSE and VaGraM models converge to a similar solution when using a neural network. This leads us to the conclusion that our loss function really only admits a single solution and that this solution coincides with the mean square error prediction when the function approximation has sufficient capacity to model the dynamics function with high precision. On the other hand, the VAML network converges to solutions that are far away from the environment sample measured in the $L_2$ norm and it cannot recover from these spurious minima due to the complex optimization landscape.

## 5 EXPERIMENT: MODEL-BASED CONTINUOUS CONTROL

Due to the limited complexity of the pendulum environment, the quantitative differences between the mean squared error and VaGraM are at times insignificant in this setting. The dynamics function of the environment can be approximated sufficiently well with a simple neural network and one hidden layer.

The underlying theory supporting VAML states that a value-aware loss is preferable to a mean squared error loss in a setting where the capacity of the model is too small to fully approximate the problem or the state space contains dimensions that are irrelevant for the control problem. To test whether our loss function is superior to a maximum likelihood approach in these cases, we used the Hopper environment from the OpenAI gym benchmark (Brockman et al., 2016). As a deep learning based Dyna algorithm, we chose Model-based Policy Optimization (MBPO) (Janner et al., 2019) and ran all of our experiments using the implementation provided by Pineda et al. (2021). We kept the structure of the MBPO algorithm and models and replaced the model loss function with VaGraM.

### 5.1 HOPPER WITH REDUCED MODEL CAPACITY

In the first experiment, we decreased the network size of the used neural network ensemble. Janner et al. (2019) use fully connected neural networks with four hidden layers and 200 neurons per layer.

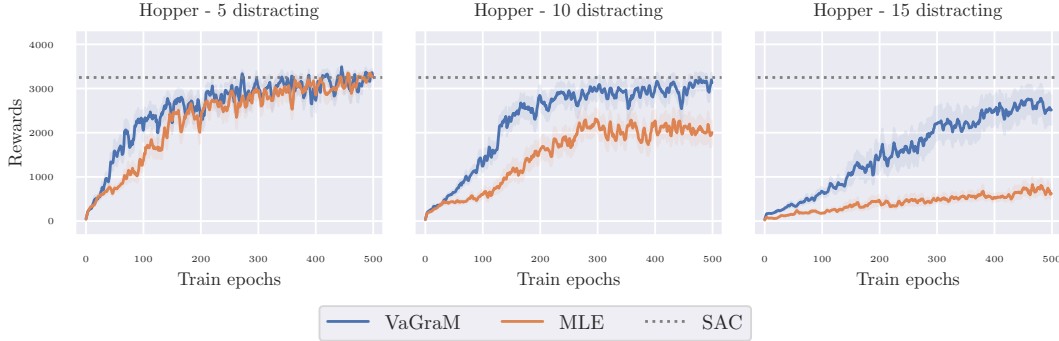

Figure 4: **Performance of VaGraM and MLE models with distracting state dimensions**. The dotted lines correspond to the final performance achieved by both algorithms on the Hopper task without distraction (grey, approx. 3200). Shaded area represents standard error over 16 repeated runs. VaGraM achieves significantly higher returns than the MLE baseline, especially in the most challenging setting with 15 distracting dimensions.

To test the performance of the algorithm under smaller models, we ran tests with two and three layer networks and 64 neurons per hidden layer. The results are shown in Figure 3. As before, when using a sufficiently powerful function approximation, we see no difference between the maximum likelihood approach and VaGraM, suggesting that the networks are flexible enough to capture the true environment's dynamics sufficiently close for planning. But when reducing the model size, the maximum likelihood models quickly lose performance, completely failing to even stabilize the Hopper for a short period in the smallest setting, while VaGraM retains almost its original performance.

## 5.2 HOPPER WITH DISTRACTING DIMENSIONS

To show that VaGraM is able to achieve good performance in a setting where there are additional dynamics in the environment that do not contribute to the control problem, we appended distractor dimensions to the Hopper observations. These are independent of the original environment state space and reward function, and evolve under non-linear and discontinuous dynamics (details in Appendix E). A setting with distracting dimensions is known to pose difficulty for model-based control algorithms (Stone et al., 2021) and neural networks struggle to model non-linear, discontinuous dynamics, so with an increasing number of dimensions the task becomes harder.

The results of this experiment are shown in Figure 4. When using five distractor dimensions, both models are able to capture the dynamics sufficiently well to achieve comparable reward to the original environment. When increasing the number of dimensions, the performance of the MLE model deteriorates, as more and more of its capacity is used to model the added dynamics. VaGraM continues to be able to achieve reward even under the presence of distracting dimensions, since the gradient of the value function with regards to the state dimensions which are irrelevant to the control problem becomes small over training. Still, the performance of VaGraM also suffers with increasing dimensions: when adding 20 distracting dimensions, neither algorithm is able to stabilize the Hopper consistently. In this case, the value function approximation cannot differentiate sufficiently between the relevant and irrelevant dimensions with the amount of environment samples provided.

In summary, we find that VaGraM is able to deal with challenging distractions and reduced model capacity significantly better than a MLE baseline. This validates that our algorithm is really value-aware and can use the value function information to improve the performance of a model-based controller in settings where the model is unable to fully represent the environment.

## 6 RELATED WORK

Several authors have noted on the problem of learning models that align with the goal of obtaining a good policy. The proposed approaches fall into three broad categories: value-function or policy dependency, representation learning, and data resampling.

Inspired by VAML, Abachi et al. (2020) present a method that seeks to align the policy gradients under a learned model with the gradients under the true environment. Similar to our proposal, D'Oro et al. (2020) also proposed to reweigh samples in a log likelihood loss, but used policy search as the reinforcement learning approach and did not account for individual state dimensions. Nikishin et al. (2022) show an approach to directly optimizing the policy performance on the real environment by learning a model with implicit differentiation. Asadi et al. (2018) show that the original VAML loss coincides with a Wasserstein distance in the model space under the assumption that the value function is Lipschitz smooth. We find that all of these approaches suffer from similar scaling issues as VAML and have not been shown to lead to strong empirical performance outside of toy settings.

Grimm et al. (2020) characterize the space of possible solutions to the model learning problem under different restrictions given by value functions and policies, and propose to learn models that are value-equivalent, similar to Farahmand et al. (2017). In a follow-up work Grimm et al. (2021) expand on this idea and show that their principle can be used to improve the MuZero algorithm (Schrittwieser et al., 2020). However, these works do not discuss the optimization challenges in actually finding such value-equivalent models, they mostly characterize the space under different value functions and policies, and present an orthogonal research direction to this paper.

An alternative to characterizing the problem in the state space of the MDP are representation learning approaches, which seek to map the original states to a latent representation that is more amenable to control. Such approaches include Value Prediction Networks (Oh et al., 2017), Embed-to-Control (Watter et al., 2015) and related proposals (Levine et al., 2020; Cui et al., 2021). Zhang et al. (2021) build on the idea of bisimulation metrics (Ferns et al., 2004; 2011) which seeks to characterize the difference between MDPs by finding a state mapping that is reward-invariant under policy and value function. In this work, we did not investigate learning state embeddings, but combining our work with representation learning approaches is an exciting direction for future research.

Lambert et al. (2020) hypothesize that the objective mismatch could be solved by reweighing the training buffer for model learning to prioritize datapoints with high value. These are more likely to matter for obtaining an optimal policy. A similar proposal was evaluated empirically by Nair et al. (2020). Contrary to our work however, this technique cannot account for the differing impact of the state space dimensions and scaling, since the data points are weighted as a whole.

## 7 CONCLUSION

We presented the Value-Gradient weighted Model loss (VaGraM), a novel loss function to train models that model a dynamics function *where it matters* for the control problem. We derived our loss function from the value-aware model learning framework, showing that previous work does not account for two important optimization phenomena that appear when learning models with empirical value function approximations. We highlighted how VaGraM counters these issues and showed the increased stability of the training procedure when using our loss in a pedagogical environment.

On the Mujoco benchmark, VaGraM performs on par with maximum likelihood estimation when using large neural network models. However, introducing additional complications to the problem results in drastic performance impacts for MLE based models, which highlights the necessity for value function aware losses in challenging environments and settings in which sufficient model capacity cannot be guaranteed. In these cases, value-awareness can greatly increase the performance of Dyna algorithms by focusing the model learning procedure on relevant aspects of the state space. In future work we seek to scale our loss function to image-based RL, where relevant state space dimensions can vary over a task due to shifting camera angles. Furthermore, we seek to derive a related value-aware approach for partially observable domains that can take the state inference problem into account.

ETHICAL CONCERNS AND LIMITATIONS

The proposed value-aware model learning approach is designed as a general purpose solution to the model mismatch problem in MBRL. While the reinforcement learning paradigm as a whole has important ethical ramifications, especially in settings where automated decision making affects humans directly, we do not address these concerns specifically in our paper as our method focuses on algorithmic problems that are orthogonal to the question of proper reward design. We restrict our proposal to cases in which the reward design actually captures the intended task, which is a common, yet rarely addressed, assumption in the RL literature.

REPRODUCIBILITY

We provide an open-source version of our code at `https://github.com/pairlab/vagram`. Furthermore we document all details on the implementation and evaluation setting in Appendix E and describe all necessary components of our loss in the main text. We also used the open source MBRL-Lib implementation for MBPO (Pineda et al., 2021) as the basis for our code and for the evaluation of baselines and provide our model loss as an additional module in the framework for easy replication.

ACKNOWLEDGMENTS

AG and AMF acknowledge the funding from the Canada CIFAR AI Chairs program, as well as the support of the Natural Sciences and Engineering Research Council of Canada (NSERC) through the Discovery Grant program. AG is also supported by the University of Toronto XSeed, LG and Huawei. We thank the members of the PAIR lab and AMF group for feedback on the paper and help with running the experiments. We are grateful to the anonymous reviewers for their constructive feedback and the valuable rebuttal discussion. Resources used in preparing this research were provided, in part, by the Province of Ontario, the Government of Canada through CIFAR, and companies sponsoring the Vector Institute.

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

## A  Bound between Value-Aware Model Learning and Value-Gradient weighted Model loss

The error in Taylor approximation $\mathcal{R}(V, s', f_\theta(s, a))$ is bounded by $\frac{M}{2}||s' - f_\theta(s, a)||^2$ with M dependent on the Hessian of the value function. Plugging this into the VAML loss and assuming worst case approximation errors, we obtain an upper bound on the VAML error:

$$\mathbb{E}_{s',s,a\sim D}\left[\left(V\left(f_\theta(s, a)\right) - V(s'_0)\right)^2\right] \tag{8}$$

$$=\mathbb{E}_{s',s,a\sim D}\left[\left((\nabla_s V(s)|_{s'})^\mathsf{T}(f_\theta(s, a) - s') + \mathcal{R}(V, s', f_\theta(s, a))\right)^2\right] \tag{9}$$

$$\leq\mathbb{E}_{s',s,a\sim D}\left[\left(|(\nabla_s V(s)|_{s'})^\mathsf{T}(f_\theta(s, a) - s')| + |\mathcal{R}(V, s'_0, f_\theta(s, a))|\right)^2\right] \tag{10}$$

$$\leq\mathbb{E}_{s',s,a\sim D}\left[\left(|(\nabla_s V(s)|_{s'})^\mathsf{T}(f_\theta(s, a) - s')| + \frac{M}{2}||s' - f_\theta(s, a)||^2\right)^2\right] \tag{11}$$

$$\leq 2 \cdot \mathbb{E}_{s',s,a\sim D}\left[\left((\nabla_s V(s)|_{s'})^\mathsf{T}(f_\theta(s, a) - s')\right)^2\right] + 2 \cdot \mathbb{E}_{s',s,a\sim D}\left[\frac{M^2}{4}||s' - f_\theta(s, a)||^4\right] \tag{12}$$

Our experiments show that if we treat $M$ like a tuneable hyperparameter, we obtain worse performance when optimizing this upper bound compared to VaGraM. The Hessian parameter is difficult to compute or estimate in practice and we find that most often, either the first or the second loss component will dominate when choosing heuristic values.

## B  Analyzing the additional local minima of the Taylor approximation loss

We noted in the main paper that the direct Taylor approximation of the value function leads to a spurious local minimum. This is clear when looking at the loss for a single datapoint:

$$\min_\theta \left((\nabla_s V(s)|_{s'})^\mathsf{T} f_\theta(s, a)\right)^2 \tag{13}$$

$$=\min_\theta \left(\sum_{n=0}^{\dim(\mathcal{S})} (\nabla_s V(s)|_{s'})_n \cdot f_\theta(s, a)_n\right)^2 \tag{14}$$

Assuming that $f$ is flexible and can predict any next state $s'$ (i.e. by choosing $f = \theta$), the optimal solution is obtained from an undetermined linear system of equations. This system admits far more solutions than either the corresponding IterVAML loss or a mean squared error, and many of them will achieve arbitrary large value prediction errors. In fact, the equation describes a hyperplane of minimal solutions consisting of every weight vector that is orthogonal to the gradient of the value function at the reference sample, with $\dim(\mathcal{S}) - 1$ free variables. Therefore we need to enforce the closeness of the model prediction and the environment sample, since the Taylor approximation is only approximately valid in a close ball around the reference sample.

One way to achieve this closeness is by adding the second order Taylor term, which results in an additional MSE loss term. As pointed out in Appendix A, we did not achieve good performance when testing out this version, since it is difficult to compute the Hessian in higher state spaces and heuristically choosing a value as a hyperparameter proved to be difficult to tune in practice. Therefore, we approached the solution to this problem as outlined in the paper.

## C  Full VAML for SAC

The formulation of VAML which we derive for SAC is a direct extension of the VAML loss to the SAC soft bellman backup. Specifically, given some distribution over the state-action space $\mathcal{D}$, state-action value function $Q$ and policy $\pi$, we can define a SAC-aware loss.

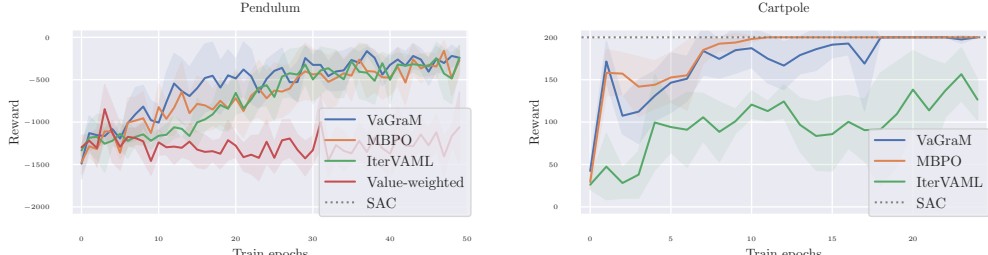

Figure 5: Comparison of VaGraM, MLE (MBPO baseline), IterVAML Farahmand (2018) and a value-weighing ablation on two simple continuous control tasks. The shaded area represents standard error estimated over 8 runs. While VaGraM, IterVAML and MBPO are able to achieve satisfactory performance on the Pendulum Swingup task, IterVAML fails to stabilize the more difficult Cartpole balancing task.

$$\mathcal{L}_{Q,\pi}(\hat{p}, p, \mathcal{D}) =$$

$$\int \left| \int (\hat{p}(s'|s,a) - p(s'|s,a))\, \mathbb{E}_{a' \sim \pi(\cdot|s')}(Q(s',a') - \log \pi(a'|s'))ds' \right|^2 d\mathcal{D}(s,a) \qquad (15)$$

$$= \int \left| \int (\hat{p}(s'|s,a) - p(s'|s,a))\, V^\pi(s',a')ds' \right.$$

$$\left. - \int (\hat{p}(s'|s,a) - p(s'|s,a))\, \mathbb{E}_{a' \sim \pi(\cdot|s')} \left[ \log \pi(a'|s') \right] ds' \right|^2 d\mathcal{D}(s,a) \qquad (16)$$

as well as its sample-based version:

$$\hat{\mathcal{L}}_{Q,\pi}(\hat{p}, p, \mathcal{D}) = \sum_i \left| V^\pi(s'_i) - \int \hat{p}(s'|s_i,a_i)V^\pi(s')ds' - \right.$$

$$\left. \left( \int \pi(a'|s'_i) \log \pi(a'|s'_i)da' - \int \int \hat{p}(s'|s_i,a_i)\pi(a'|s') \log \pi(a'|s')da'ds' \right) \right|^2 \qquad (17)$$

We find that in most of our experiments, the entropy terms did not significantly contribute to the loss. Therefore we dropped it in our experiments and directly derived VaGraM for the value function prediction error only. This makes our loss applicable in all situations were the model is used to estimate the value function of a policy.

Finally, we do not account for the dependency of the policy function update on the model. Since SAC aims to minimize the KL divergence between the action distribution and the Gibbs distribution defined by the value function directly, the only way the model influences this update is via the induced state space distribution. We do not address this matching explicitly in this work, but a hybrid policy-aware and value-aware model is an enticing direction for future research.

## D    ADDITIONAL EXPERIMENTS

### D.1    ABLATIONS

To test the performance of VaGraM and MBPO against alternative models, we used the classic Pendulum swingup and Cartpole benchmarks. As points of comparison, we used IterVAML (Farahmand, 2018) and a simple value-weighted regression similar to Nair et al. (2020), where the MSE error is multiplied by the inverse of the value function of the sample. The results are visualized in Figure 5. Hyperparameters follow the Cartpole task baseline in Pineda et al. (2021).

Since IterVAML suffers from strong destabilization on the Pendulum model learning problem, as discussed in section 4, we added an MSE loss to the original formulation $\mathcal{L}_{\text{joint}} = \mathcal{L}_{\text{IterVAML}} + \lambda \mathcal{L}_{\text{MSE}}$

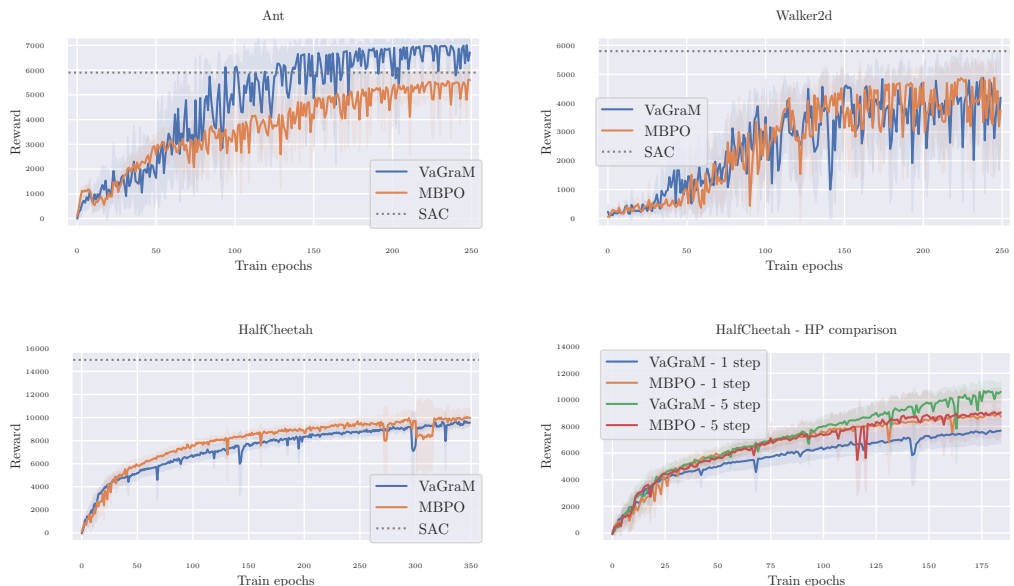

Figure 6: Comparison of VaGraM and MBPO on the Mujoco tasks presented in Janner et al. (2019). We see that VaGraM is able to perform on par with the MBPO implementation on all tasks, while outperforming both the model-based and model-free baseline on the Ant environment. For the HalfCHeetah-v2 task, different performances emerge when using different rollout lengths. While the MLE baseline is not able to benefit from longer rollouts, VaGraM increases in performance, although the statistical significance of this increase is low due to the strong bimodality of solutions on the HalfCheetah task.

with a tradeoff factor of $\lambda = 0.01$. We see that this is sufficient to stabilize the performance in the Pendulum swingup task and prevent catastrophic divergence, but the model still performs significantly below the MBPO and VaGraM results in the Cartpole stabilization task. This is evidence that even with additional stabilization mechanisms, the issues presented in section 3 prevents the easy adoption of IterVAML to complex domains.

The value-weighing baseline is unable to achieve satisfactory performance even on the simple pendulum task, leading us to conclude that more complex reweighing schemes such as VaGraM are indeed necessary.

## D.2 PERFORMANCE IN MUJOCO BENCHMARK SUITE

The comparison runs on the Mujoco benchmark environments presented in Figure 6 are faulty due to a bug that was pointed out to us at the poster session. The bug only harms the performance of VaGraM, so we do not see it as a reason to withdraw the paper, but due to limited time and computational resources, we were unable to repeat the full comparison presented here. All other experiments are corrected and to the best of our knowledge reflect the performance of VaGraM. We will update the paper with updated comparisons once they are finished.

The results of VaGraM and MBPO are presented in Figure 6. We find that VaGraM is able to perform on par with the MBPO baseline on all the tasks. Due to stability issues, the Humanoid-v2 task is excluded, no algorithm achieved satisfactory performance (compare Pineda et al. (2021) for a discussion).

On the Ant-v2, we see small performance improvements above the results reported by Janner et al. (2019) and Pineda et al. (2021). We hypothesise that for this task, the canonical state space representation could be misaligned with the control problem in the sense that not all dimensions are informative for control. A further investigation of the phenomenon would be an interesting direction

for future research. All comparisons were done on the hyperparameter settings provided by Pineda et al. (2021).

Investigating the performance differences further, we find that on the HalfCheetah-v2 domain, Va-GraM is able to profit from longer rollouts, while MBPO does not increase in performance when rolling out the model for more than 1 step. The comparison is presented in Figure 6. The MLE based model completely destabilizes if trained beyond 180 - 200 epochs on longer rollouts, which is why we truncated the training length before convergence. The performance of VaGraM with longer rollouts is already beyond the final performance of the fully trained MBPO baseline after 180 steps.

The destabilization leads to a complete collapse of the policy performance, probably due to increasing uncertainty of the model and in some cases to a gradient explosion that produces NaN values in the policy. We find that VaGraM does not suffer from this pattern and is able to slightly outperform both MBPO and VaGraM trained on a single step rollout. Similar patterns do not emerge in the Walker-v2 domain, here training on longer rollouts destabilizes all algorithms. We leave a more in-depth discussion of the impact of rollout length on policy performance under different models and losses for future work but highlight that significant performance might be gained by finding better tradeoffs than those discussed in Janner et al. (2019), especially when comparing deterministic and probabilistic models (compare Appendix F) as well as value-aware and value-agnostic models.

## E    EXPERIMENT DETAILS

As described in the main paper, we conducted our experiment in two domains from the OpenAI benchmark OpenAI Gym (Brockman et al., 2016), Pendulum-v2 and Hopper-v0. We used both environments as provided by the framework without further modification in our capacity tests. Our algorithm is shown in pseudocode in algorithm 1.

**Expanded Hopper environment**    To test the performance of VaGraM in a setting with additional unnecessary state observations, we created a random dynamical system that evolves according to randomly initialized dynamics function, where $A$ is a fixed matrix with entries randomly drawn from $\mathcal{N}(0., 10.)$, and a fixed initial state $s_0$ with components randomly drawn from $\mathcal{N}(0., 0.1)$. We do not resample the fixed components at reset.

The transitions are then described by the following deterministic function:

$$f(s_t, a_t) = \begin{cases} s_t + \sin(As) & \text{if } |f(s_t, a_t)| < 20 \\ s_0 & \text{else} \end{cases} \tag{18}$$

This dynamics function contains two attributes which are hard for neural networks to model: discontinuity and non-linear dynamics. We find that this is sufficient to provides a very challenging environment for the MLE trained neural network models. To account for varying difficulty over different random initialization of the environments, we made sure to test the comparison runs with the same seeds over the different loss functions, but we did not find that the inter-seed variance was larger than the observed difference between the different loss functions.

### E.1    ARCHITECTURE AND HYPERPARAMETERS

For the Pendulum experiments, we use a simple fully connected neural network with a single layer, and a linear regression without feature transformations as architectures. The used non-linearity is ReLU. All experiments were implemented in PyTorch (Paszke et al., 2019) and were not specified, the standard initialization of the library were kept. The exact versions of all used libraries are documented in the provided source code.

To assure a fair comparison we used the hyperparameters provided by Janner et al. (2019) for all experiments with our approach and the NLL loss function used for the baseline. All models used were fully connected neural networks with SiLU non-linearities and standard initialization. For all experiments with full model size we followed MBPO and used seven ensemble members with four layers and 200 neurons per layer.

---

**Algorithm 1:** Value-Gradient weighted Model learning (VaGraM)

---

Initialize policy $\pi_\phi$, value function $v_\psi$, model $\hat{f}_\theta$, environment dataset $\mathcal{D}_{\text{env}}$, model dataset $\mathcal{D}_{\text{model}}$;

**for** *N epochs* **do**

    **while** $\hat{p}_\theta$ *not converged* **do**

        Sample batch $(s, a, r, s')$ from $\mathcal{D}_{\text{env}}$;

        $\mathcal{L}_{v_\psi} = \left( (s' - \hat{f}_\theta(s,a))^\intercal \text{diag} \left( \frac{d}{ds} v_\psi(s)|_{s'} \right)^2 (s' - \hat{f}_\theta(s,a)) \right)$;

        $\theta \leftarrow \theta - \alpha \frac{d}{d\theta} \mathcal{L}_{v_\psi}$;

        Train reward model

    **end**

    **for** *E steps* **do**

        Take action in env according to $\pi_\phi$; add to $\mathcal{D}_{\text{env}}$;

        **for** *M model rollouts* **do**

            Sample $s$ from $\mathcal{D}_{\text{env}}$; sample $a \sim \pi_\phi(s)$;

            $s', r = \hat{f}_\theta(s,a)$;

            Add $(s, a, r, s')$ to $\mathcal{D}_{\text{model}}$

        **end**

        **for** *G policy gradient updates* **do**

            Sample batch $(s, a, r, s')$ from $\mathcal{D}_{\text{env}} \cup \mathcal{D}_{\text{model}}$;

            $\psi \leftarrow \psi - \beta \frac{d}{d\psi} (v_\psi(s) - (r + \gamma v'(s')))^2$;

            $\phi \leftarrow \phi - \lambda \hat{\nabla}_\phi J(\pi_\phi, v_\psi, (s, a, r, s'))$

        **end**

    **end**

**end**

---

Even though our loss derivation does not support rolling out the model for more than a single step without accounting for this in the training setup, we find that VaGraM is still stable over the short rollout horizons used by MBPO. Therefore we used the rollout scheme from MBPO.

However, it was necessary to make a small alteration to the training setup: in the provided implementation, the value function is solely trained on model samples. Since our model is directly dependent on the value function, we need to break the inter-dependency between model and value function in the early training iterations. Hence, we used both real environment data and model data to train the value function, linearly increasing the amount of model samples from 0 to 95% of the SAC replay buffer over the first 40 epochs of training (corresponding to 40.000 real environment steps). We did not transition to fully using model data to assure that the real environment samples are still able to inform the value function learning. We found that this did not diminish the training returns of MBPO compared to solely using model samples and so used this approach for both VaGraM and MLE.

To estimate the gradient of the value function, we used the four empirical value functions, two direct estimates and two target value functions used in the SAC algorithm and summed the losses using each data tuple and value function gradient independently. Furthermore we calculated the $L_2$ norm of all value function gradients and clipped these at the 95-th percentile. We found that this was necessary since the empirical value function gradients can get very sharp, which in rare cases leads to a destabilization of the gradient descent algorithm used to update the model.

## F    DETERMINISTIC VS PROBABILISTIC MODELS

Our derivation of VaGraM lead us to the conclusion that a deterministic model was sufficient to achieve the goal of value-aware model learning in environments with small transition noise. This insight stands in contrast to the current literature, which often claims that probabilistic models are needed to achieve optimal performance in model-based reinforcement learning. However, there is no clear consensus among different authors whether probabilistic models are needed or if a deterministic model can be sufficient for MBRL (compare Lutter et al. (2021)).

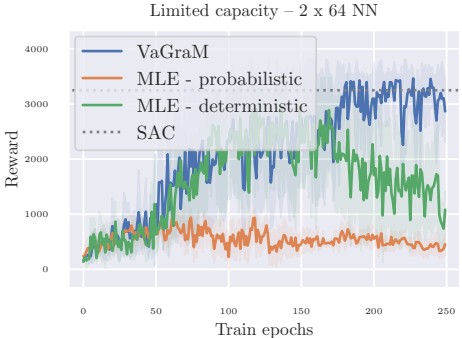 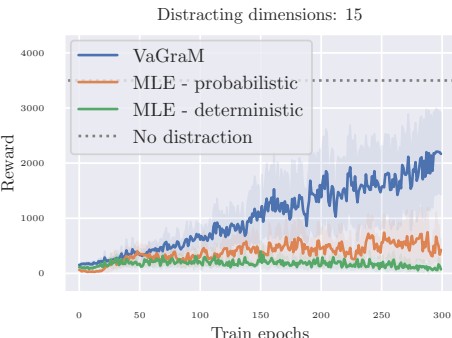

Figure 7: **Comparison of the empirical performance of deterministic and probabilistic MLE models vs VaGraM**. Thick lines denote the mean and shaded area the standard error over 8 runs. In the limited capacity setting, the deterministic model is able to achieve significantly higher returns than the probabilistic baseline, however with stability issues during longer training. On the distracting benchmark, the deterministic model is not able to achieve any significant returns.

Our assumption that the model can be replaced with a deterministic one relies on the assumption that the underlying model is not dominated by stochastic transitions, but is deterministic or near deterministic and unimodal. This follows from the requirement that the model prediction admits a small error in the mean squared error sense, otherwise the Taylor approximation does not properly capture the behavior of the value function, as a model prediction might have high likelihood under the environment and still be far away from the environment sample in a mean squared sense otherwise. In domains where capturing the stochasticity is crucial, we have to revisit this requirement in follow up work.

Furthermore, we are operating under the assumption that the mean value function of each distribution over states can be represented as the value function of a single state prediction. Since value function approximations represented by neural networks are continuous, this is true in our setting due to the mean value theorem for integrals. Implicitly due to the Taylor approximation, we also assume that this mean value lies close to or on the environment sample we obtained. If other approximation schemes are used, this property needs to be checked and potentially probabilistic models are needed to represent the expectation over all possible value functions and transition dynamics.

### F.1 ABLATION EXPERIMENT WITH DETERMINISTIC MODELS

In our experiments in the Hopper domain, we used probabilistic models following Janner et al. (2019). To verify that the improvement in performance did not result from using a deterministic model instead of a probabilistic model, we repeated the experiment showing the impact of smaller model sizes, and replaced the probabilistic Gaussian ensemble with an ensemble of deterministic functions trained with the mean squared error loss between sample and state prediction. All other implementation details, architecture and hyperparameters were kept fixed. The results are shown in Figure 7. In this ablation, we do indeed see that a mean squared trained model capture the dynamics information better than a model that is trained using negative log likelihood. To the best of our knowledge, this phenomenon has not received attention in the literature, but we hypothesize that it is an artifact of training a probabilistic model with gradient descent instead of natural gradient descent (compare Figure 1 in Peters & Schaal (2008) for a visual intuition).

Nonetheless, VaGraM is still able to achieve higher cumulative reward consistently than the MSE model. Especially on the distraction task we find that a deterministic MSE model performs on par with the probabilistic model and fails to achieve any reward when faced with a challenging number of distractions. This validates our hypothesis that value awareness is important in settings with insufficient model capacity, but crucial in cases where the environment observations are not aligned with the control problem.

