# OpenReview forum: "Value Gradient weighted Model-Based Reinforcement Learning"
_ICLR.cc/2022/Conference — ICLR 2022 Spotlight_

### Official Review · Reviewer_dE85 · 2021-10-28

**Correctness:** 3
**Technical Novelty And Significance:** 3
**Empirical Novelty And Significance:** 3
**Recommendation:** 8
**Confidence:** 5

**Main Review:**

## General Comments
- **writing quality**: This is a well written paper. Throughout, it was easy to follow where the authors are going and why.
- **technical novelty**: The VaGraM method seems well motivated. The related works and introduction of the objective mismatch issue are well done. The mathematics are easy to follow throughout.

A question I had / an intuition that I am curious how the authors would respond to: when training dynamics models, in my experience the models are pretty accurate on average and the error follows a normal distribution (with some points being very inaccurate), do you think your loss function and resulting algorithms would be able to handle these outlier error points? I think the Taylor approx. could get weird with big prediction errors?

- I looked at the code and am familiar with the library the authors use. Everything they mention seems to be there, but I did not take the time to run all the experiments.
- **experimental validation**: Due to a lot of other things the authors had to explain, there isn't a lot of room for experiments in this paper. Generally, the experiments support the authors claims, but some of them are somewhat questionable motivated (ie adding the irrelevant states in state-based MBRL). I've added some more questions on the experiments below, as they form the central way for the authors to improve the score of the paper.

exper.1: the proposed "challenges" for MBRL do not seem to match up with the literature. Can the authors comment more on the motivation for small model capacity and distracting state dimensions?

exper.2: small but crucial change -- it is near impossible to see the VaGraM line in figure 2. I saw it by zooming in a ton, but printing the paper would render it invisible. Please add markers to this and preferably all plots.

exper.3: I think of a better test for models handling distracting states is to try it on high-dimensional tasks like humanoid. The paper the authors references (Stone et al 2021) focuses on pixel-based RL, which seems very different than what the authors are doing. Does this problem come up more in the literature? If the authors showed their loss function outperforming MBPO on a task like humanoid that would make me fight hard for its acceptance.

More detailed questions on the experiments are below, but these are the main points.

## Comments By Section

### 1. Intro
- 1.1 do the authors have any citations for discontinuities being hard to model? It seems like something that is folk-wisdom but unproven?
- 1.2 I really like the intro of the paper, good job authors. I would potentially add a hint that VaGraM uses value function gradients in the abstract. Now it just says value-aware learning. Maybe that's just preference!
- 1.3 I think the last sentence of the contributions is oversold. At least in the paper and in a limited read of the appendix, the experiments are a little limited in breadth to say it is on par with existing algorithms. It's on par on a couple tasks with one algorithm. It is certainly promising, but that didn't quite match with the experiments.

### 2. Background
- 2.1 only comment is that I think the sum across i in equations 1,2, and on could be better described. Is that the sum across the state-dimension?

### 3.Value-Gradient Weighted Model Loss
- 3.1 The core issue for me in this section is figure 1. Either it seems like the VAML loss matches the shape / the color scales for higher is better / something is making the results hard to read. Are all the learning solutions showing loss that is too high around the area of interest? Why is value lower outside? Is there a simpler explanation to all of this - I don't see the VaGraM method being better as described in the text.
- 3.2 is there a dropped negative sign in equation 4? Otherwise, how does the subtracted integral become positive? I see that one of the terms cancels out.
- 3.3 in section 3.2 how is this bound implemented? Is this just a numerical bound on what the loss function can achieve or an implementation trick? That wasn't totally clear to me. Is there an experiment in the appendix showing the effects?
- 3.4 what constitutes a "small prediction error" for the Taylor expansion to hold? Any examples to give on a simulated task?
- 3.5 the authors speaking about dynamics models "as there is no guarantee to which local optimum the algorithm will converge" is there any reason to think that the "optimum model" wrt to loss will be helpful, in the light of objective mismatch?
- 3.6 can the authors clarify footnote 2, this was confusing to me.

### 4. Example: Cartpole
- 4.1 AVI is used but not defined.
- 4.2 Is the source of the value function across domains important? Ie the difference between IterVAML and SAC for generating it?
- 4.3 I just wanted to confirm that the dataset generated and environment transition function used to generate next samples is to get the true value function for pendulum? End of second to last paragraph p6.
- 4.4 the authors say "a model with insufficient capacity to represent the dynamics." Isn't the standard pendulum task started vertical, then the dynamics are expressible by a linear function as long as the system is pretty stable?

### 5 High-dimensional
- 5.1 if the cited paper is for pixel based RL? What is the motivation here? The result is interesting, though it seems fairly limited in scope when it is easy to make these relatively small models bigger?
- 5.2 the change by layer including a smaller layer size of 64 neurons is weird. What happens with only changing the amount of layers and having a width of 200 for all (4,3,2 layers)? As said above, what happens with the algorithm on Humanoid?

## Nits / grammatical comments
- n1 the abstract on open review has latex commands that didn't work, I am not sure if you can fix them :) \algoNameFull (\algoName)
- n2 end of page 5, "this , it" is missing a word
- n3 in non-generalizing local minima section, this reads weird "Even worse, there examples where..."
- n4 in section 2.1 MBRL review,  the idea of an "unknown true MDP's transition" is a little weird. I get it, but non-experts may be confused

## Other comments / ideas (not included in scoring of paper)
- o1 I appreciate the author's comment on the ethical concerns regarding reward hacking and RL.
- o2 I would be very interested in how the
- o3 are the claims regarding "Single solution convergence" true across seeds? What would this look like? Is there interesting insight to be gained there?


**Summary Of The Paper:**

The authors propose a new loss function for training dynamics models in MBRL that is aware of the environmental value. This loss function is a tool to mitigate the "objective mismatch" issue and seems well motivated. The authors show the usefulness of their VaGraM in simple experiments.


**Summary Of The Review:**

A well written and motivated paper needs a bit of clarification on the motivation and execution of experiments. The motivation makes me lean towards accept, but I would appreciate if the authors fixed or clarified some of the limitations.

---

> ### Author Response · Authors · 2021-11-14
> **Answer to review**
>
> **Detailed Answer**
>
> Thank you for your detailed and encouraging review. We are happy to see that you think our contributions are well written and motivated!
>
> The main concern is that the empirical evaluation of the algorithm could be improved., We concur with this assessment and will add the missing mujoco benchmarks (Ant, Walker, HalfCheeath and Cartpole) to the appendix. The humanoid benchmark suffers from stability issues and has not been reproduced with a Pytorch implementation of MBPO, therefore we forgo a comparison on it. We will also add IterVAML as a baseline to our Hopper experiments.
>
> Regarding the validity of our experiments which were discussed in several places in the review, we think a longer answer is valuable. This specifically addresses points exper 1 and 5.1:
>
> The commonly expressed challenges for MBRL are model error and, related, model mismatch. The settings in which model error occurs are, to the best of our knowledge, insufficient model capacity or insufficient state space coverage by data. Since the second problem is inherent to RL and difficult to solve without making strong assumptions on the environment, we focus on the first. While rarely addressed explicitly in deep learning literature, model capacity problems and learning challenges are a fact that every algorithm has to contend with, so we do not think that this is a particularly contentious claim.
>
> Distracting state dimensions have been pointed out as a challenge mostly in the visual RL setting, but we do not see a reason why testing their impact in the state-based setting should be different or less interesting. While it is commonly assumed that state spaces for control problems are hand crafted to provide good information for obtaining good policies, in more complex or natural systems such an assumption might not be valid. Consider e.g. a medical scenario where doctors don’t know a priori which physiological data is informative for a treatment. Or an autonomous vehicle, in which a pre-trained visual system gives object information about various entities in the visual field of the vehicle, not all of which might be necessary for good driving.
>
> The main reason why we do not test our hypothesis in the pixel-based setting is due to the additional orthogonal complexity in these settings. Our paper focuses on presenting the core idea and simple experiments where we can highlight the qualitative and quantitative impact of changes in the loss. Scaling the algorithm up to pixel-based observations requires overcoming additional difficulties which we want to tackle in future work but which would be out of scope for this paper.
>
> While making models bigger in our experiments is indeed easy, we present our evaluation as a stand-in for a case where this is not easily achievable. Network size is relative to problem complexity, and so while it is easy to come up with a task with high dimensionality that is hard to solve without large networks, this incurs heavy training costs which seems unreasonable for the proof of our concept. We therefore mimic the setting of insufficient model size by artificially constraining model sizes on a simpler benchmark. This allows us to show the presence and solution of the model mismatch without having to contend with additional challenges in larger settings such as exploration or training cost.

---

> > ### Author Response · Authors · 2021-11-14
> > **Detailed answers (part 1)**
> >
> > > “would be able to handle these outlier error points? I think the Taylor approx. could get weird with big prediction errors?”
> >
> > This is a valid concern, although since we show how our loss relates to MSE, we do not think that it behaves significantly “weirder” than MSE. The biggest problem occurs if the local gradient of a point is very small but the curvature is large, so adjacent points have a far larger gradient. In this case our loss will erroneously decrease the error incurred by a wrong prediction. This is easily fixed by adding a constant offset to the main diagonal of our scaling matrix (equivalent to adding the MSE error), but we have not found this to be crucial in our experiments.
> >
> > > “exper.2: small but crucial change -- it is near impossible to see the VaGraM line in figure 2. I saw it by zooming in a ton, but printing the paper would render it invisible. Please add markers to this and preferably all plots.”
> >
> > We will improve the figure as suggested.
> >
> > > “1.2 I really like the intro of the paper, good job authors. I would potentially add a hint that VaGraM uses value function gradients in the abstract. Now it just says value-aware learning. Maybe that's just preference!”
> >
> > Thanks for pointing this out, this is indeed preferable!
> >
> > > “2.1 only comment is that I think the sum across i in equations 1,2, and on could be better described. Is that the sum across the state-dimension?”
> >
> > We have clarified this in the update draft, it is the sum over the data points in the training set/training batch.
> >
> > > “3.1 The core issue for me in this section is figure 1. Either it seems like the VAML loss matches the shape / the color scales for higher is better / something is making the results hard to read. Are all the learning solutions showing loss that is too high around the area of interest? Why is value lower outside? Is there a simpler explanation to all of this - I don't see the VaGraM method being better as described in the text.”
> >
> > The main point of Figure 1 is to show that VAML has a very complicated loss, while MSE is a simple square and VaGraM is a combination of both that aligns with the local gradient of the value function on the reference point along the coordinate system axis. Note that the colors do not signify which one is better, only how large the incurred loss will be if the model predicts point (x,y) instead of the marked reference point. Furthermore the shading of each plot is relative, since the scaling of the losses is different. This is also why we have not provided a scale, the loss magnitude across different formulations is not directly comparable. The “better” claim arises from the fact that VaGraM follows the shape of the value function, while MSE does not.
> >
> > We agree that the figure is slightly difficult to interpret, even though it gives important intuition for the discussed loss functions. We have expanded the caption accordingly:
> >
> > _Visualization of discussed loss function with regards to a reference point marked with the white cross and the corresponding value function on the Pendulum environment. For the value function, darker color indicates a lower value. In the loss figures, darker color indicates how large the loss is if the model predicts $(x,y)$ instead of the reference sample marked in white. The VAML loss has a complex non-linear shape in the state space that follows isolines of the value function, while MSE and \algoName are centered around the sample. For VaGraM, the rescaling of the MSE in the direction of high gradient along the $\theta$ axis is visible. Due to the upper bound introduced in (7), the scaling is aligned with the axis of the coordinate system and not rotated to fit the value function closer_
> >
> > > “3.2 is there a dropped negative sign in equation 4? Otherwise, how does the subtracted integral become positive? I see that one of the terms cancels out.”
> >
> > The negative sign is dropped due to the square. As you noted, the value function itself cancels and the remaining term is negative and squared, so we dropped the negative.
> >
> > > “Is this just a numerical bound on what the loss function can achieve or an implementation trick? That wasn't totally clear to me. Is there an experiment in the appendix showing the effects?”
> >
> > This is a numerical bound and not an implementation trick. We will add a comparison of the “pure” Taylor expansion and our upper bound to the appendix which makes the difference clear.
> >
> > > “3.4 what constitutes a "small prediction error" for the Taylor expansion to hold? Any examples to give on a simulated task?”
> >
> > The error of the Taylor expansion is in $O(\nabla^2 f(x)|_x (x - x’) ^2)$, which we also discussed in the appendix. It is difficult to concretely say how small the error can be for a given task, since this depends on the value function and its higher order derivatives. We find that in most of our experiments, the model is useful for learning good control which we see as an indirect validation that the model error is indeed small.

---

> > > ### Author Response · Authors · 2021-11-14
> > > **Detailed answers (part 2)**
> > >
> > > > “3.5 the authors speaking about dynamics models "as there is no guarantee to which local optimum the algorithm will converge" is there any reason to think that the "optimum model" wrt to loss will be helpful, in the light of objective mismatch?”
> > >
> > > The general performance of MBRL can be bounded by the total variational distance between the model and the true environment or the VAML loss as shown in 2.2 paragraph 3. Therefore convergence to a model which minimizes either of these quantities provides at least an upper bound on the error. In the deep learning setting it is hard to qualify whether the specific local optimum found over either loss will be a tighter performance bound than the other, but since VAML is a tighter bound on the performance difference between model and environment, we think it is reasonable to assume that minimizing this loss will be preferable in general.
> > >
> > > > “3.6 can the authors clarify footnote 2, this was confusing to me.”
> > >
> > > We have updated footnote 2 to read: _The full loss function will likely still admit additional local minima due to the non-linear nature of the model itself, but the global optimum will coincide with the true model_
> > >
> > > > “4.1 AVI is used but not defined.”
> > > We have added the full term and a reference.
> > >
> > > > “4.2 Is the source of the value function across domains important? Ie the difference between IterVAML and SAC for generating it?”
> > > Is the question referring to the difference between AVI and SAC? There is indeed a difference since SAC’s value function includes an entropy regularization term that depends on the policy. We discuss the difference in more detail in the appendix, but we found this additional term to be negligible in our experiments.
> > >
> > > > “4.3 I just wanted to confirm that the dataset generated and environment transition function used to generate next samples is to get the true value function for pendulum? End of second to last paragraph p6.”
> > >
> > > The dataset is only used for the model training and is constructed to assure that we have proper state space coverage. The value function is trained in the regular fashion by updating the policy and taking exploratory steps in an online RL loop. We could have also used the artificial dataset to create an even more precise value function estimate, but we wanted the value function used to be similar to what could be expected from a regular online RL loop.
> > >
> > > >“4.4 the authors say "a model with insufficient capacity to represent the dynamics." Isn't the standard pendulum task started vertical, then the dynamics are expressible by a linear function as long as the system is pretty stable?”
> > >
> > > We used the Pendulum Swingup task, in which a Pendulum is first swung up and then stabilized. The swingup can not be linearized well enough. We agree that “Inverted Pendulum” more commonly refers to stabilizing an upright pendulum on a cart and have changed the text to clarify.
> > >
> > > > “5.1 if the cited paper is for pixel based RL? What is the motivation here? The result is interesting, though it seems fairly limited in scope when it is easy to make these relatively small models bigger?”
> > >
> > > The cited paper is only used as an example to show that RL suffers in the presence of distracting dimensions. Most state space RL environments are handcrafted to only contain relevant information so we added these artificially, but even in the original Mujoco benchmarks, we see that not all dimensions are equally important for prediction. See also our comment to the feedback on the experiments.
> > >
> > > > “5.2 the change by layer including a smaller layer size of 64 neurons is weird. What happens with only changing the amount of layers and having a width of 200 for all (4,3,2 layers)?”
> > >
> > > MBPO is surprisingly robust to reducing the number of layers, we found that reducing their size was a far more challenging comparison. We are also rerunning a full comparison on NNs with 2, 3, and 4 layers and 64 and 200 neurons; we will include these in the appendix.
> > > Overall we found it difficult to estimate what changes to the networks would reduce their capacity since the literature on the theoretical learning capabilities of MLPs is only developing.

---

> > > > ### Comment · Reviewer_dE85 · 2021-11-15
> > > > **Brief comment**
> > > >
> > > > RE the last point on MBPO - that is an interesting contribution imo. Adding a brief phrasing of what you just mentioned to the main text would be something I like as a reader.

---

> > > ### Comment · Reviewer_dE85 · 2021-11-15
> > > **Responses / discussion**
> > >
> > > RE "This is a valid concern, although since we show how our loss relates to MSE, we do not think that it behaves significantly “weirder” than MSE."
> > >
> > > This is interesting. It's likely out of scope by being way too detailed, but I would look at any further ablations trying to understand this. E.g. with a grid of different densities.
> > >
> > > RE figure 1. The caption update is very useful, thanks.
> > >
> > > Eqn 4: ah, got it.
> > >
> > > Great clarifications here.

---

> > ### Comment · Reviewer_dE85 · 2021-11-15
> > **Response to discussion of experiments**
> >
> > Your points are reasonable. I appreciate the detailed response to how the distracting states and model dimension impact MBRL. I think a lot of these directions are under-explored (or not at all).
> >
> > ### Distracting states
> > Personally, I think distracting states may matter more in pixel-based due to the way that the background interacts with the foreground -- in state-based, distracting states will always be the same index (pixel based may come and go as the agent moves). That being said, it is certainly worth exploring, I just don't yet think it will be a huge impact.
> >
> > ### additional experiments
> > Yes, these are very useful. MBPO seems to have some re-implementation issues, so adding more results that could figure out "why" is very useful. A very good addition.

---

> ### Comment · Reviewer_dE85 · 2021-11-15
> **Reviewer Responses**
>
> The reviewers did a very good job updating the paper timely and engaging in positive discussion. In light of the edits done now and edits promised for the final version, I have updated my recommendation to accept.

---

### Official Review · Reviewer_kWH6 · 2021-11-01

**Correctness:** 3
**Technical Novelty And Significance:** 3
**Empirical Novelty And Significance:** 4
**Recommendation:** 6
**Confidence:** 5

**Main Review:**

## Strengths:

1. The paper is well-written. It has a clear motivation for improving upon the VAML model loss empirically and presents two important shortcomings of the VAML objective. The second issue on unstable local minimas is also highlighted empirically in the Inverted Pendulum environment.

2. The proposed VaGraM loss is designed to specifically remedy the two presented issues in the original VAML loss. The derivation is sound and the final loss function is intuitively appealing.

3. In the InvertedPendulum environment, two valuable empirical insights are presented: (a) Qualitatively (Figure 1), the proposed loss landscape is shown to be similar to MSE in it’s parabolic or “single-minima” nature, while also being sensitive to the value function landscape. This is in contrast to VAML whose loss landscape does not admit a single optimum. (b) In a controlled setting where value function and policy is selected at various stages of training of a model-free approach for the purpose of learning a model independently (with periodic value updates), the VAML loss is shown to converge to minimas that are unstable and cause the VAML loss to diverge significantly once the value network is updated (particularly in the NN - 1 hidden layer case). This supports the earlier presented hypothesis about the second shortcoming of VAML.

4. In the complex continuous control MuJoCo domain, the Hopper environment is used to present two key benefits of the proposed objective over MLE: (a) it retains performance with reduced model capacity and (b) it is comparatively more robust to MLE with increasing distractor dimensions in the state space. The latter experiment supports the main motivation for using VAML -- the fact that it is theoretically supposed to ignore parts (or dimensions) in the state space that are irrelevant for the value function.


## Weaknesses

1. The empirical analysis in the paper leaves a lot to be desired for gaining a thorough understanding of the benefits of VaGraM.
For the complex continuous control environments, why was Hopper the only environment selected? It would be good to see the performance of the proposed model loss in other MuJoCo environments as well, especially HalfCheetah, Walker and Ant in which the open-source MBPO implementation by Pineda et al. (2021) is shown to work well. It is okay to skip Humanoid because they haven’t gotten it to work, but the others should be included. These environments need not be used for all the experiments on model-capacity and distracting dims -- just showing a benchmark of reward curves on each unmodified environment would be sufficient.

2. Why is the VAML baseline not present in the Hopper experiments in Section 5? Even if it performs poorly, including it is important for the sake of completeness (as the earlier InvertedPendulum experiments do include the VAML baseline).

3. The experiments on the Inverted Pendulum environment provide valuable insights. However, they do not include a comparison of the proposed method vs MLE and VAML trained end-to-end in an MBRL algorithm (i.e. the opposite of the controlled experimental setting where the policy, value and model all interact and influence each other). Such an experiment has been presented for the more difficult Hopper environment (left-most plot of Figure 3), but it is not presented for the easier InvertedPendulum setting.

4. In the InvertedPendulum experiment represented by Figure 2, Tte paper shows that with the selected value update frequency, the VAML error diverges in the NN setting. I have two questions regarding this observation -- (a) Why not include a baseline that performs gradient clipping to prevent the exploding losses in VAML?, (b) Why not vary the value update frequency from once every 1000 model updates to a different value (possibly in conjunction with gradient clipping from my previous point), to test whether the VAML loss still diverges with higher value update frequency? Given these potential remedies, the presented second issue with VAML (“Non-generalizing local minima”) may actually be a non-issue, i.e. it may be an artificial problem that can be overcome in a simpler manner.

5. Given that the taylor expansion in Eqn 4 only uses a linear approximation, this approximation is only accurate when consecutive states (s and s’) in the environment are close to each other. This assumption is in addition to the assumption presented in the paper that the model prediction should not be too different from the environment’s next state. It seems that this requirement for consecutive states to be close to each other is naturally met in most continuous state space environments such as the ones chosen in this paper. However, it may be the case that other environments have sharp state space transitions (or discrete state spaces) where the linear approximation may no longer hold. The paper should comment on the requirement of this assumption and when it may be violated, and whether or not it holds in the selected environments.

## Factual Errors/Corrections

1. The conclusion mentions “MuJoCo benchmark” -- this is not accurate as only the Hopper environment was selected.
Section 5.2, last line: “improve the model learning performance” -- Here, ‘model learning performance’ is not defined. If it refers to the policy performance on Hopper as a result of the proposed model learning loss, then that is not really improved in the original unmodified architecture and state space setting, so this may be misleading. If it refers to actual performance of the model, then what is the metric? Is it the VAML error?

1. Section 5 title: “High-Dimensional…” -- The selected Hopper environment isn’t high dimensional. This is misleading as it suggests that the environments use high dimensional state spaces such as RGB images as opposed to the joint and position information of the simulated MuJoCo robot.

## Other Issues/Comments:

* Figure 1: The “Ours” plot has a darker region that does not seem to be aligned with the value function axis in the left-most “Value function” plot. Is this an intended consequence or is there some error in this plot?



**Summary Of The Paper:**

The paper proposes a new loss for model-learning in model-based RL, called “Value-Gradient weighted Model loss” (VaGraM). The loss belongs to the family of value-aware model learning losses and is inspired by the VAML and IterVAML losses by Farahmand et al. The paper designs this loss with two main issues of VAML in mind -- (1) value function being evaluated on irrelevant model predictions and (2) model converges to optimas that are sensitive to value function perturbation, being de-stabilized easily when value function is slightly updated. The proposed method is shown to fix both of these issues and is studied in the pedagogical Inverted Pendulum environment and in the MuJoCo Hopper environment. The key takeaways from the Hopper experiments is that the proposed method does better than the baseline MLE model loss in two scenarios -- (1) reduced model capacity and (2) distracting dimensions.

**Summary Of The Review:**

The motivation and presentation of the proposed method are major strengths of the paper. The experiments presented on the simple (InvertedPendulum) and complex (Hopper) environments are insightful.

The paper has potential to be a strong contribution if the empirical analysis can be expanded based on the weaknesses I had mentioned in my review.

---

> ### Author Response · Authors · 2021-11-14
> **Answer to review**
>
> **Detailed Answer**
>
> Thank you for your detailed and encouraging review. We are happy to see that you think our contributions are insightful and the paper can be a strong contribution!
>
> The main concern is that the empirical evaluation of the algorithm could be improved., We concur with this assessment and will add the missing mujoco benchmarks (Ant, Walker, HalfCheeath and Cartpole) to the appendix. The humanoid benchmark suffers from stability issues and has not been reproduced with a  Pytorch implementation of MBPO, therefore we forgo a comparison on it. We will also add IterVAML as a baseline to our Hopper experiments.
>
> **Specific comments**
>
> > “Why is the VAML baseline not present in the Hopper experiments in Section 5?”
>
> We were unable to find a training regime in which IterVAML shows any performance gains over a random baseline and have excluded it due to this. Since this question came up in several reviews, we will add the corresponding curve.
>
> > “The experiments on the Inverted Pendulum environment provide valuable insights. However, they do not include a comparison of the proposed method vs MLE and VAML trained end-to-end in an MBRL algorithm”
>
> We omitted this comparison since there is no clear benefit of using VAML over MLE in this setting as the dynamics are very easy to model even with small networks. To clarify this, we will add the experiment to the appendix.
>
> > “it may be an artificial problem that can be overcome in a simpler manner.”
>
> We spent considerable time tuning IterVAML hyperparameters and had to conclude that this problem is indeed not due to a simple hyperparameter or training setting, but an inherent problem of the proposed algorithm in IterVAML. It can be overcome by adding an MSE loss term to IterVAML, but an analysis of the gradient shows that in almost all cases one of the two terms will dominate in the testing environments and the performance is suboptimal to using simple MSE (compare [Decision-Aware Model Learning for Actor-Critic Methods: When Theory Does Not Meet Practice](https://proceedings.mlr.press/v137/lovatto20a.html) ).
>
> Training the model for fewer steps (i.e. early stopping) or clipping the updates only delays the divergence, since the model can still move towards local minima outside the state space region covered by the data. In addition, finding an optimal tradeoff between model accuracy and early stopping in more complex environments has proven to be very difficult, since the problem only becomes apparent *after* updating the value function.
>
> > “Given that the taylor expansion in Eqn 4 only uses a linear approximation, this approximation is only accurate when consecutive states (s and s’) in the environment are close to each other”
>
> There seems to be a misunderstanding: the Taylor expansion is around the _next state_ sampled from the replay buffer (compare equation (3)); we do not make any assumptions on the difference between s and s’. Such an assumption would be important to estimate the difficulty of learning f(s,a), which we do not address in this work.
>
> Our assumption will hold in near-deterministic MDPs and sufficiently powerful function approximators, which was also commented on by reviewer 2. In addition, it is similar to the assumptions necesary for MSE regression. We added two clarifications to 3.1 that explain this assumption and the implications.
>
> Added:
> _We also assume that the model has small transition noise, akin to the model assumptions underlying MSE regression, otherwise the difference between a model sample and the next state sample might be large._

---

> > ### Author Response · Authors · 2021-11-14
> > **Answer part 2**
> >
> > > “Figure 1: The “Ours” plot has a darker region that does not seem to be aligned with the value function axis in the left-most “Value function” plot. Is this an intended consequence or is there some error in this plot?”
> >
> > Due to our upper bound in section 3.2, the loss is not rotationally aligned with the value function, but aligned with the state space coordinate system. We can prevent this by taking the second order Taylor expansion, but at significant additional computational cost ($O(d^2)$ where d is the dimensionality of the state space). Furthermore the shading of each plot is relative, since the scaling of the losses is different. This is also why we have not provided a scale, the loss magnitude across different formulations is not directly comparable. Since other reviewers also had questions on how to interpret the plot, we have expanded on the explanation provided in the caption.
> >
> > New caption:
> >
> > _Figure 1:
> > Visualization of discussed loss function with regards to a reference point marked with the white cross and the corresponding value function on the Pendulum environment. For the value function, darker color indicates a lower value. In the loss figures, darker color indicates how large the loss is if the model predicts $(x,y)$ instead of the reference sample marked in white. The VAML loss has a complex non-linear shape in the state space that follows isolines of the value function, while MSE and VaGraM are centered around the sample. For VaGraM, the rescaling of the MSE in the direction of high gradient along the $\theta$ axis is visible. Due to the upper bound introduced in (7), the scaling is aligned with the axis of the coordinate system and not rotated to fit the value function closer_
> >
> > > “The conclusion mentions “MuJoCo benchmark” -- this is not accurate as only the Hopper environment was selected.”
> >
> > You are correct, we are adding the remaining benchmarks and will update the text accordingly after all comparisons are done.
> >
> > > “If it refers to the policy performance on Hopper as a result of the proposed model learning loss, then that is not really improved in the original unmodified architecture and state space setting, so this may be misleading.”
> >
> > We agree that it is possible to misunderstand the claim. We updated the text accordingly.
> >
> > Changed:
> > _This validates that our algorithm is really value-aware and can use the value function information to improve the performance of a model-based controller in settings where the model is unable to fully represent the environment._
> >
> > > “Section 5 title: “High-Dimensional…” -- The selected Hopper environment isn’t high dimensional.”
> >
> > While the term “high-dimensional” is relative and we were using it in reference to IterVAML and the previous experiments, it is indeed more common to use it to refer to pixel observations. We have changed the title of the section correspondingly to “Model-based Continuous Control”

---

### Official Review · Reviewer_WgYP · 2021-11-02

**Correctness:** 4
**Technical Novelty And Significance:** 4
**Empirical Novelty And Significance:** 2
**Recommendation:** 8
**Confidence:** 5

**Main Review:**

Strengths

- The paper is novel. It isolates the existing problems with VAML and proposes a loss that 1) has a unique optimum avoiding the shortcomings of VAML 2) addresses the objective mismatch of MLE-based algorithms, thus opening the possibility to get the best of both worlds.
- MBRL algorithms that address the objective mismatch can learn the dynamics that are helpful for policy improvement but produce unrealistic samples. The VaGraM loss is an MLE weighted by sensitivity of value function to states. As a result, the learned model produces samples that are close to real yet focuses on components that are important for control.
- The empirical part of the paper seems trustworthy: the practical implementation builds on top of standard baselines, the experiments perform controlled studies, and the results are averaged over a sufficient number of runs.
- The paper is generally well written and easy to follow.

Weaknesses

- The experiments miss a few relevant baselines. Since the VaGraM loss is value-gradient-weighted MLE, natural baselines would be 1) return-weighted MLE 2) the method from [1] that uses policy-gradient-weighted MLE. While the reviewer appreciates that the authors mention [1] in related work and discuss that VaGraM is value-based and [1] is policy-based, the quality of the paper would be improved if the experiments had these or other related benchmarks that also address the objective mismatch.
- The significance of the results could be increased by adding more environments. Right now, the continuous control experiments on MuJoCo are limited to only Hopper. Reporting the results on ~5 other tasks (like the baseline MBPO) would demonstrate whether the conclusions generalize beyond a single task.

Detailed comments and questions

- One of the motivations of the paper is that training models using value estimates outside of the empirical state distribution might result in inadequate VAML losses. While the proposed loss indeed removes the dependency on V(s), it still depends on $\nabla_s V(s)$. Hence, VaGraM might perform inadequately when $\nabla_s V(s)$ is evaluated outside of the state distribution. On the other hand, it is unclear whether it is tractable to ask for guarantees outside of the training distribution. Do you have any thoughts on that?
- Figure 1 (right) shows the rescaling effect of VaGraM on the loss landspace. However, in addition to scaling, the value estimates in Figure 1 (left) are rotated. Would it be possible to extend the method with the curvature information e.g. using Hessians instead of $\textrm{diag}(\nabla_s V(s))$
- Is it correct that on Figure 2 (right) MSE and VaGraM losses coincide? If so, that’s very interesting, but the reviewer is unsure whether it’s desirable or not. Could you elaborate on the observation?

[1] D'Oro, Pierluca, Alberto Maria Metelli, Andrea Tirinzoni, Matteo Papini, and Marcello Restelli. "Gradient-aware model-based policy search." In Proceedings of the AAAI Conference on Artificial Intelligence, vol. 34, no. 04, pp. 3801-3808. 2020.

---------------------------------------------------------------

POST-REBUTTAL UPDATE:

I increase the score to 8. The authors have addressed my major concerns. See the discussion for details.

**Summary Of The Paper:**

The paper studies the model learning aspect in model-based reinforcement learning (MBRL). Standard Dyna-based MBRL algorithms rely on maximum likelihood estimation (MLE) or its variants for model learning. In contrast, the authors propose a Value-Gradient weighted Model (VaGraM) loss. Using the loss, the authors mitigate the objective mismatch in standard MBRL: models with high likelihood might not be good for maximizing returns since they optimize a different objective. The paper builds on top of value-aware model learning (VAML) but points out two problems with the original algorithm: 1) value function estimates can be arbitrary outside of the empirical state distribution 2) the entanglement of value and model learning creates the possibility for bad local optima. The experimental results provide evidence that the proposed VaGraM loss alleviates the problems and outperforms in the MLE baseline when the true model cannot be estimated accurately.

**Summary Of The Review:**

The reviewer leans towards recommending the submission for acceptance. However, while the idea and the insights are sound, the experimental results could be strengthened by adding relevant baselines and testing on more environments. Addressing the outlined concerns might increase the overall score.

---

> ### Author Response · Authors · 2021-11-14
> **Answer to review**
>
> **Detailed Answer**
>
> Thank you for your detailed and encouraging review! We are happy that you agree that our work is interesting and well presented!
>
> The core concern is that the empirical evaluation of the algorithm could be improved. We concur with this assessment and will add the missing mujoco benchmarks (Ant, Walker, HalfCheeath and Cartpole) to the appendix. The humanoid benchmark suffers from stability issues and has not been reproduced with a  Pytorch implementation of MBPO, therefore we forgo a comparison on it.
>
> We will also try to compare to [1]. There is no open source implementation of their work, but we have already contacted the authors regarding access to the code
>
> **Specific comments**
>
> > “The experiments miss a few relevant baselines. Since the VaGraM loss is value-gradient-weighted MLE, natural baselines would be 1) return-weighted MLE 2) the method from [1] that uses policy-gradient-weighted MLE.”
>
> Thank you for suggesting these additional baselines. We are unsure how they would fit in our framework, since reward-weighted MSE could be seen as an additional algorithm that we do not discuss in our paper. We will try to perform a comparison, but if there is a specific paper which implements such an algorithm, we would be thankful if it were pointed out so that we have a more solid basis for comparison.
>
> > “While the proposed loss indeed removes the dependency on V(s), it still depends on [...]”:
>
> The model loss does indeed still depend on the gradient of the value function, but crucially this gradient is only evaluated during training on s’ sampled from the training data. This means that the value function is explicitly trained on this data point.
> We added a clarification to 3.1 after equation (4) that explains the difference more thoroughly.
>
> Changed:
> _This objective function crucially does not depend on the value function at the model prediction which solves the first of our major problems with the VAML paradigm_
>
> to
>
> _[...] does not depend on unknown state samples, the value function is trained on all $s'$ in the dataset [...]_
>
>
> > “Figure 1 (right) shows the rescaling effect of VaGraM on the loss landspace. However, in addition to scaling, the value estimates in Figure 1 (left) are rotated. Would it be possible to extend the method with the curvature information e.g. using Hessians instead of [...]”
>
> Yes, this is possible, but we did not do this due to the computational cost of evaluating the Hessian during training, as it scales quadratically with the dimensionality of the state space. A short discussion can be found in the appendix.
>
> > “Is it correct that on Figure 2 (right) MSE and VaGraM losses coincide? If so, that’s very interesting, but the reviewer is unsure whether it’s desirable or not. Could you elaborate on the observation?”
>
> This is indeed the case and we agree that it is an interesting observation. Our hypothesis is that since our loss can be interpreted as a reweighted MSE, the solutions of the algorithms will coincide with sufficient model capacity and data coverage (barring convergence to suboptimal local minima which might be different for both losses).
> We added this clarification to the figure caption and will also update the figure to show both loss curves better.
>
> New figure caption:
>
> Figure 2:
> _Evolution of the VAML loss over changing value functions on the Pendulum domain}. Lines denote the mean and shaded areas show standard error over 8 model initialization and data set samples per model. In the linear setting, VAML achieves the lowest VAML error, while \algoName is able to significantly outperform MSE. In the NN setting, VAML diverges rapidly, while VaGraM and MSE converge to approximately the same solution._

---

> > ### Comment · Reviewer_WgYP · 2021-11-18
> > **Follow-up discussion**
> >
> > Dear Authors,
> >
> > Thank you for the reply and for the clarifying details.
> >
> > Since the discussion period when authors can update the draft ends in a couple of days, I am curious whether you have some preliminary results on the Ant, Walker, HalfCheeath, and Cartpole environments? If so, do you observe the same benefits of VaGraM on experiments with distractors and reduced model size? Is training of VaGraM stable on other environments?
> >
> > Regarding the return-weighted MLE, I believe it can be seen as a policy-gradient-like loss but for model learning. However, since the loss is quite simple, I am unsure if there is a paper that proposes this loss.

---

> > > ### Author Response · Authors · 2021-11-18
> > > **Follow-up**
> > >
> > > Dear reviewer,
> > >
> > > We are finalizing the experiments, and have updated the draft.
> > >
> > > We spent substantial time on trying to make Humanoid-v2 possible following the wishes of reviewer [de85](https://openreview.net/forum?id=4-D6CZkRXxI&noteId=ic3HTSOrA-O) but were unable to overcome the problems reported by Pineda et al.
> > >
> > > We find that the loss is stable for the Ant, Walker, HalfCheeath, and Cartpole environments and is able to outperform MBPO on Ant with the original hyperparameters. We also added comparisons to IterVAML and a Value-weighted baseline on the Pendulum Swingup and Cartpole task like you suggested. We did not repeat the additional experiments with distracting dimensions and reduced model size because a lot of tuning is involved in making sure that the environments do not become too difficult for all algorithms to solve. Investing in better benchmarks for robustness in MBRL which are not only focused on visual observations is one of the take-aways from this project. But given the strong results on Ant, I believe our hypothesis is supported beyond the environments previously presented.
> > >
> > > We do not fully understand what a policy-gradient like loss would look like for model learning. Do you suggest just weighing the MSE by the reward or value? As a simple baseline/ablation we added a loss suggested by [Nair et al.](https://proceedings.mlr.press/v119/nair20a/nair20a.pdf) (Section 3.4) $\mathcal{L} = \\sum_{i=0}^N \frac{1}{|V(s'_i)|} (s'_i - f(s_i,a_i)) ^2$ which only weights the squared error by the value. This did not turn out to be a strong alternative, so we did not extend it to further environments. We also tested weighing by the absolute value of the value function instead of the inverse like Nair et al. suggested, but to similar effect.
> > >
> > > The authors of d'Oro provided us with their code but it was not in a shape that allowed an easy comparison on our environments.

---

> > > > ### Comment · Reviewer_WgYP · 2021-11-19
> > > > **Continued discussion**
> > > >
> > > > Dear Authors,
> > > >
> > > > Thank you for adding the additional experiments. I increase the score to 8 and confidence to 5 and will be recommending the paper to be accepted.
> > > >
> > > > I strongly encourage you to invest more time in further experimentation, e.g. reporting the performance with distractors / limited network size on more environments. It will significantly increase the impact and visibility of the paper.
> > > >
> > > > Regarding the return-weighted baseline, I was thinking about something like that:
> > > > $L(\theta) = \sum_{i=1}^N \||s'_i - f_\theta(s_i, a_i)\||^2 G_i$, where $G_i$ is a Monte-Carlo estimate of the return associated with a state-action pair $(s_i, a_i)$. Maybe there are other simple alternatives...

---

> > > > > ### Author Response · Authors · 2021-11-19
> > > > > **Continued discussion**
> > > > >
> > > > > Thanks for your suggestions and updating the scores.
> > > > >
> > > > > Regarding the proposal, our new baseline on the Pendulum Swingup task performs essentially this update where we set $$G_i = \hat{V}(s'_i)$$, amortizing over the Monte Carlo return estimate using the value function. This also makes intuitive sense since the scale of the value function can be correlated with model errors, but as our comparison shows, this does not seem to turn out to be true in practice. We think that rewards are not the correct quantity to weigh by because in sparse reward settings, we would expect the algorithm to wrongly attribute importance only to states adjacent to the goal.

---

### Official Review · Reviewer_VsMY · 2021-11-03

**Correctness:** 4
**Technical Novelty And Significance:** 3
**Empirical Novelty And Significance:** 2
**Recommendation:** 6
**Confidence:** 5

**Main Review:**

Strengths:
- The objective and the work is clearly motivated and grounded in previous work.
- Paper is generally well written and easy to follow.
Weaknesses:
- The proposed objective is limited to deterministic dynamics; a more details discussion of this limitation is preferred in the main paper as opposed to the appendix.
- The paper mainly lacks experimental evidence —
    - how does VaGram fare compared to VAML[1] and other value-aware techniques (e.g. Modhe et al. [2])?
    - do the trends in hopper hold in other mujoco environments?
    - How does the proposed technique compare against state-of-the-art MBRL / MFRL methods? (Note that I do not expect outperforming them. But, a comparison will help better understand the contributions in better context).

[1] Farahmand, Amir-massoud. "Iterative Value-Aware Model Learning." In NeurIPS, pp. 9090-9101. 2018.
[2] Modhe, Nirbhay, Harish Kamath, Dhruv Batra, and Ashwin Kalyan. "Model-Advantage Optimization for Model-Based Reinforcement Learning." arXiv preprint arXiv:2106.14080 (2021).

**Summary Of The Paper:**

The paper introduces a new value-aware objective for model learning that directly builds on VAML — specifically, using a Taylor series approximation for the next state value prediction w.r.t the state under consideration. This yields a somewhat intuitive form upon simplification — essentially, dimension-wise weighted L2 distance between predicted and actual states of the model where the weights correspond to the gradient. The experimental results in restricted domains indicate that the proposed VaGram method outperforms MLE.

**Summary Of The Review:**

The proposed objective is straight-forward and can be of use to the MBRL community. However, I do think adding additional experiments to put the contribution in better context will make the work more useful for members of the community -- in both, understanding the significance of the work and potentially, improving value-aware objectives for MBRL.

---

> ### Author Response · Authors · 2021-11-14
> **Answer to review**
>
> **Detailed Answer**
>
> Thank you for your detailed and encouraging review! We are happy that you agree that our work is interesting and well presented!
>
> The main concern is that the empirical evaluation of the algorithm could be improved. We concur with this assessment and are in the process of adding the suggested mujoco benchmarks (Ant, Walker, HalfCheeath and Cartpole) to the appendix.  Moreover, we will also add IterVAML as a baseline to our Hopper experiments. Notably, the humanoid benchmark suffers from stability issues and has not been reproduced with a Pytorch implementation of MBPO, therefore we forgo a comparison on it.
>
> We will also try to compare to [2]. There is no open source implementation of their work, but we have already contacted the authors regarding access to the code. We will do our best to perform a comparison if we get access in time, but we cannot promise a fair comparison without the code.
>
> **Specific comments:**
>
> > “The proposed objective is limited to deterministic dynamics...”
>
> Our model can account for some small homoscedastic noise in the transition function when the value function curvature is small, similarly to other MSE regression approaches. Nonetheless, we agree that this assumption should be presented more clearly in the text of the main paper. We have stated this assumption in 3.1 paragraph 3 and will add further clarification when this assumption is valid.
>
> Added:
> _We also assume that the model has small transition noise, akin to the model assumptions underlying MSE regression, otherwise the difference between a model sample and the next state sample might be large._
>
>
> > “How does the proposed technique compare against state-of-the-art MBRL / MFRL methods?”
>
> To the best of our knowledge, MBPO remains the state of the art method in state-based continuous control. Several more recent papers either fail to clearly outperform MBPO or are tailored towards slightly different problems such as RL from pixels. Note that our algorithm currently does not apply to POMDP settings or settings with additional representation learning. We plan to extend it to these settings in future work.
> If there are specific algorithms which we can compare against to add weight to our claims, we are happy to include further comparisons.

---

### Author Response · Authors · 2021-11-14
**General answer to reviews**

We thank the reviewers for their encouraging and thorough reviews! We will answer most points in individual replies and offer a brief summary here.

We are happy to see that all reviewers agree that the work is clearly motivated, novel and well written. The reviewers also agree that the core idea of reweighing learning objective is interesting and the experiments presented in the paper show promise. Moreover, all the reviewers point out that the ideas are presented in an accessible fashion and any suggestions on minor presentation fixes in the figures are now improved in the updated version.

A common concern is that the experimental evaluation of our method can be improved by including more comparisons on a broader set of environments. We would like to underline that the main aim of our paper is to highlight the impact of the model loss under different exemplary challenges, which is undoubtedly achieved by the current experiments. Nonetheless, we will add the remaining OpenAI gym mujoco environments as well as a comparison to “IterVAML” as suggested by [reviewer VsMY](https://openreview.net/forum?id=4-D6CZkRXxI&noteId=McnzxXiEFAo) and the ablations suggested by [reviewer WgYP](https://openreview.net/forum?id=4-D6CZkRXxI&noteId=4DvoaSdmQ5F).

For the additional suggested baselines (d’Oro et al. and Modhe et al.), no publicly available implementations exist, and it would be impractical to get a working version implemented and tested without extensive time investment. We will, however, do our best to make a fair comparison. We have contacted the respective authors and will include comparisons if we get access to reasonable code, but we cannot perform this fairly without tested and reliable implementations.

Since many of the requested experiments are computationally intensive some of the comparisons are still running. We will add all additional experiments to the paper during the week and will notify the reviewers and Area Chairs when the draft is updated.

**Note**: We included suggested improvements in our updated draft and prepared a latexdiff so that all reviewers can quickly see the differences. The tool has slightly changed the formatting of the paper, we will revert this before the end of the rebuttal.

---

### Author Response · Authors · 2021-11-18
**Updated draft**

We updated the draft to the paper (changes in blue).

**The following changes were made:**

**Writing** We updated _Non-generalizing local minima_ in Section 3 to make it clear the the issue appears when updating the value function and spurious local minima in the state space exist (not the parameter space).

**Experiments** We added Pendulum, Cartpole, Ant, Walker and HalfCheetah results to the appendix. We also added IterVAML as suggested by reviewer [VsMY](https://openreview.net/forum?id=4-D6CZkRXxI&noteId=4DvoaSdmQ5F) to the simple Cartpole and Pendulum task. We also implemented stabilization on VAML to address the concerns raised by reviewer [kWH6](https://openreview.net/forum?id=4-D6CZkRXxI&noteId=Y7I3QprUOB2) to show that it is indeed possible to alleviate the presented difficulties to some degree with tricks, but VAML still destabilizes quickly with slightly more difficult tasks (here moving from Pendulum Swingup to Cartpole), which validates our result that more principled solutions are needed.

**Results** We find that VaGraM is stable and on par with MLE in the Mujoco tasks, and outperforms MBPO and the model-free SAC baseline used by Janner et al. in the Ant-v2 experiment (we do not claim to be able to outperform SAC in principle, different implementations have shown performance difference between 6000 and 7000 points average return on Ant). We also find that some hyperparameter tuning is possible to increase the performance of VaGraM on HalfCheetah.

**Mistakes** We noticed a figure was missing from Appendix F which verifies that the improvement on the distracting dimensions environment is not due to introducing a deterministic transition model. We have added the relevant figure.

---

### Author Response · Authors · 2021-11-23
**Final update**

We have updated the draft. Only spelling errors and formatting was fixed.

We thank the reviewers for their continued discussion of the draft. If there are any open questions after the draft update period, we are happy to clarify.

---

### Public Comment · ~Claas_A_Voelcker1 · 2023-06-16
**Updated version**

In the published version of the paper, some experiments in the appendix are affected by a bug that was pointed out to us after submission. We updated the paper on Arxiv with an updated appendix, please refer to that version for future reference: https://arxiv.org/abs/2204.01464

---

### Decision · Program_Chairs · 2022-01-20

**Decision:**

Accept (Spotlight)

**Comment:**

This paper studies model-based RL in the setting where the model can be misspecified. In this case, MLE of model parameters is a not necessarily a good idea because the error in the model estimate compounds when the model is used for planning. The authors solve this problem by optimizing a novel objective, which takes the quality of the next state prediction into account.

This paper studies an important problem and this was recognized by all reviewers. Its initial reviews were positive and improved to 8, 8, 6, and 6 after the rebuttal. The rebuttal was comprehensive and exemplary. For instance, one concern of this paper was limited empirical evaluation. The authors added 5 new benchmarks and also included stabilizing improvements in their original baselines. I strongly support acceptance of this paper.